# New Species of *Didymellaceae* within Aquatic Plants from Southwestern China

**DOI:** 10.3390/jof9070761

**Published:** 2023-07-19

**Authors:** Tong Chen, Siyuan Wang, Xinwei Jiang, Ying Huang, Minghe Mo, Zefen Yu

**Affiliations:** 1Laboratory for Conservation and Utilization of Bio-Resources, Key Laboratory for Microbial Resources of the Ministry of Education, Yunnan University, Kunming 650091, China; chentongynu@163.com (T.C.); wsy0526lld@163.com (S.W.); jiangxinwei@mail.ynu.edu.cn (X.J.); yinghuang@ynu.edu.cn (Y.H.); 2School of Life Sciences, Yunnan University, Kunming 650091, China

**Keywords:** aquatic plant, endophyte, multi-locus phylogeny, taxonomy

## Abstract

Members of *Didymellaceae* have a wide geographical distribution throughout different ecosystems, and most species are associated with fruit, leaf, stem and root diseases of land plants. However, species that occur in aquatic plants are not clearly known. During a survey of the diversity of endophytes in aquatic plants in Yunnan, Sichuan, and Guizhou provinces, we obtained 51 isolates belonging to *Didymellaceae* based on internal transcribed spacer region (ITS) sequences. Further, the phylogenetic positions of these isolates were determined by combined sequences composed of ITS, partial large subunit nrRNA gene (28S nrDNA; LSU), RNA polymerase II second largest subunit (*rpb*2) and partial beta-tubulin gene (*tub2*). Combining morphological characteristics and multi-locus phylogenetic analyses, two new varieties belong to *Boeremia* and 12 new species distributed into seven genera were recognized from 51 isolates, i.e., *Cumuliphoma*, *Didymella, Dimorphoma*, *Ectophoma*, *Leptosphaerulina, Remotididymella*, and *Stagonosporopsis.* Among these species, only one species of *Stagonosporopsis* and two species of *Leptosphaerulina* show teleomorphic stages on OA, but have no anamorphic state. Each new species is described in detail, and the differences between new species and their phylogenetically related species are discussed here. The high frequency of new species indicates that aquatic plants may be a special ecological niche which highly promotes species differentiation. At the same time, the frequent occurrence of new species may indicate the need for extensive investigation of fungal resources in those aquatic environments where fungal diversity may be underestimated.

## 1. Introduction

Aquatic plants are the plant groups that remain submerged or float on the surface of the water [1]. Based on the different habitats and physiological characteristics during their life cycle, aquatic plants are generally classified into five types: emergent plants, floating-leaved plants, free-floating plants, submerged plants, and wet plants [2,3]. Most aquatic plants are well adapted to their surroundings, and they play an important role in maintaining the normal functioning of aquatic ecosystems [4]. Aquatic plants play an important role in maintaining water quality, including removing excess nutrient loads, absorbing nutrient mineral ions and reducing sediment resuspension [5,6].

Fungal endophytes refer to the kinds of fungi that live within plant tissues (roots, stems and leaves, etc.) without causing disease in plants [7]. They play an active role in plant growth by protecting the host plant from pests and pathogens and improving the adaptability of plants to extreme environments [8,9,10]. In addition, many studies have reported that fungal endophytes frequently produce diverse active metabolites that contribute to the well-being of the host plant [11]. Some endophytes can be symbiotic or beneficial in one case, such as providing resistance to hosts against different biotic and abiotic stresses, but potential pathogens in another, such as the barley pathogen *Ramularia collo-cygni* [12,13]. Most of the aquatic plant endophytes play critical roles in nutrient absorption and nitrogen fixation and have antioxidant and antimicrobial capacity, which may contribute to plants’ stress resistance to aquatic environments [14,15,16]. Therefore, fungal endophytes are receiving increasing attention as an important source of microbial biocontrol products. As of now, endophytes have been investigated in temperate, tropical and subtropical terrestrial plants [17,18,19,20,21]. However, little attention has been paid to plants in freshwater ecosystems. The study of endophytes in aquatic plants will inevitably lead to the discovery of fungal resources that are different from those in terrestrial environments, which will be of great importance both for the study of fungal species diversity and the development of novel secondary metabolites.

*Didymellaceae* was established by de Gruyter et al. [22] as the largest family in the *Pleosporales* (*Ascomycota*, *Pezizomycotina*, *Dothideomycetes*), encompassing three main genera, namely *Ascochyta*, *Didymella* and *Phoma* [23]. The family is distributed in a wide range of hosts and habitats. Most species are pathogens of plants, causing damage to leaves and stems and fruit rot [24,25], while other species are endophytes, saprophytes and opportunists in soil, water, air and nematodes, and even in some extreme environments such as glaciers and deep-sea sediments [25,26,27,28,29,30]. However, the classification of *Didymellaceae* has proven to be challenging. Their substrate and morphological characteristics have been used to distinguish *Didymellaceae* since the 1870s. For example, based on differences in conidial septa and sporulation patterns, Boerema and Bollen distinguished *Phoma* from *Ascochyta* [31]. The classifications based on morphological characters have been shown to be artificial and do not correctly reflect the phylogenetic status of this group of fungi [26,27,32]. Previous research also investigated *Didymellaceae* based on their morphology and phylogenetic analyses, but it did not distinguish any closely related taxa within several species complexes [33]. Therefore, the phylogenetic positions of *Didymellaceae* were subsequently revised based on morphological studies and multiple gene sequences obtained from four loci, including the internal transcribed spacer region (ITS) sequences, partial large subunit nrRNA gene (28S nrDNA; LSU), RNA polymerase II second largest subunit (*rpb*2) and partial beta-tubulin gene (*tub2*) [24,34]. The approach combining molecular phylogenetic analysis with morphological characters can effectively improve the scientific classification of species. In the last revision of *Didymellaceae*, 44 genera were accepted, and more than 5400 species have been recorded by MycoBank [25,35,36,37]. 

Southwestern China has high species diversity and is known as one of the world’s 34 biodiversity hotspots [38,39]. The variable climate, complex topography, and other abiotic and biotic factors provide a refuge for glacial-age organisms, contributing to this area harboring an inestimable diversity of fungi [38,40]. Our project investigated the diversity of endophytic fungi in roots, stems, and leaves of aquatic plants in southwestern China, and sought to explore the environmental factors that influence the fungal diversity, including geographical location, host, etc. During this process, 1689 strains were isolated, among which 51 isolates belonged to *Didymellaceae* based on the ITS sequences [41], and 51 isolates were further studied combining four loci, including ITS, LSU, *rpb*2 and *tub*2. Finally, 12 new species, two new varieties and 14 known species were recognized. Of these species, sexual morphs of three new species were observed on OA. Our results revealed the distribution of *Didymellaceae* in aquatic environments and its relationship with aquatic plants, which may help to combine endophytes and aquatic plants to improve water pollution.

## 2. Materials and Methods

### 2.1. Collection of Aquatic Plant Samples

From 2015 to 2016, aquatic plants distributed into 32 species belonging to 16 families were collected from Yunnan, Guizhou and Sichuan provinces, including 16 sample sites in Yunnan, four sample sites in Guizhou, and 11 sample sites in Sichuan. Detailed plant lists, locations and characteristics of sampling sites are provided in our previous publication [41]. At each sampling site, dominant plants were collected. For each species, at least 15 individuals were collected, with at least 5–10 m between any two individuals. Healthy and mature plants with undamaged leaves were completely uprooted and cleaned, then complete plant samples were transported back to the laboratory and stored at 4 °C.

### 2.2. Isolation of Endophytic Fungi

After the aquatic plant samples were thoroughly washed with tap water, each plant sample was cut into 20–30 mm, and subsequently, the plant samples were processed in the following order: First, the samples were placed in sterile distilled water (30 s), then immersed in 0.5% sodium hypochlorite (2 min), then in sterile distilled water (1 min), then in 75% ethanol (2 min), and finally again in sterile distilled water (1 min) [42]. After surface sterilization, plant tissues (roots, stems and leaves) were cut into small pieces of approximately 5 mm length and placed uniformly in Petri dishes containing Rose Bengal Agar (RBA; Guangdong Huan Kai Microbial Technology Co., Ltd., Guangzhou, China). Samples were incubated at 25 °C and observed for growth periodically. When the colonies grew to a level suitable for isolation and identification, the mycelium was picked onto a new plate containing potato dextrose agar (PDA; 200 g potato, 20 g dextrose, 18 g agar, 1000 mL distilled water) for purification.

The pure culture and dried culture specimens of this study were stored in the Herbarium of the Laboratory for Conservation and Utilization of Bio-Resources, Yunnan University, Kunming, Yunnan, China (YMF).

### 2.3. Morphological Characterization

Agar plugs removed from fresh culture on PDA were incubated on oatmeal agar (OA; 40 g oatmeal, 18 g agar, 1000 mL distilled water), malt extract agar (MEA; 30 g malt powder, 3 g peptone, 18 g agar, 1000 mL distilled water) and PDA at 25 °C to induce sporulation. Colony diameter and culture traits were determined at 7 days and 14 days of incubation, respectively. Colony colors were assessed according to the charts of Rayner [43]. Micromorphological structures of mature ascomata/conidiomata, ascospores/conidia and conidiogenous cells from OA cultures were placed in sterile distilled water for observation. Observation was performed using a light microscope (Olympus BX51, Japan) under differential interference contrast (DIC) illumination, using an Olympus DP controller (v.3,1,1208) software of the Olympus DP 10 digital camera to capture. The average measurements of the structures were calculated from 30 measurements.

### 2.4. DNA Extraction, Amplification and Sequence Analysis

Total genomic DNA was extracted from fresh mycelia according to the protocol described by Turner et al. [44]. Four genetic fragments of *Didymellaceae* strains, namely, LSU, ITS, *rpb*2, and *tub*2, were amplified by using the following primer pairs: LR0R and LR7 for LSU [45,46], RPB2-5F2 and RPB2-7cR for *rpb*2 [47,48], ITS4 and ITS5 for ITS [49,50], and Btub2Fd and Btub4Rd for *tub*2 [33]. The PCRs were performed in the PCR thermal cycle programs described by Chung et al. [51] in a total volume of 25 µL. The PCR mixture consisted of 1 µL of DNA template, 1 µL of forwarding primer (10 µM), 1 µL of reverse primer (10 µM), 12.5 µL T5 Super PCR Mix(containing Taq polymerase, dNTP, and Mg^2+^, Beijing TsingKe Biotech Co., Ltd., Beijing, China), 5 µL of PCR buffer, and 4.5 µL of sterile water. The entire PCR reaction was processed using an Eppendorf Mastercycler (Eppendorf, Hamburg, Germany).

All PCR products were tested by 1.5% agarose gel electrophoresis, and then purified using a commercial kit (Bioteke Biotechnology, Wuxi, China) according to the manufacturer’s instructions. The purified PCR products were sequenced forward and reverse using a LI-COR 4000 L automated sequencer and Thermo Sequenase-kit as described by Kindermann et al. [52]. Raw sequence chromatograms were checked manually, and then each isolate was compared using ClustalW in MEGA 6.06 to generate concordant sequences [53]. The consensus sequences were deposited in the GenBank database at the National Center for Biotechnology Information (NCBI). The reference sequences and new sequences generated in this study are listed in Appendix A.

### 2.5. Sequence Alignment and Phylogenetic Analysis

The phylogenetic tree was constructed with four loci, ITS, LSU, *rpb*2 and *tub*2, to identify these isolates of *Didymellaceae* at the species level. The reference strains were selected on the basis of the high sequence similarity of BLAST searches of ITS in GenBank and the adjacent strains provided by recent studies of *Didymellaceae* [54,55,56,57,58,59]. We selected 310 isolates that had been sequenced and deposited in GenBank as reference strains, referring to 34 representative genera. All sequences used in this study are listed in Appendix A. The generated sequences were manually aligned with CLUSTAL_X v. 1.83 [60] with default parameters. Aligned sequences of multiple loci were concatenated and manually adjusted through BioEdit v. 7.0.4.1 [61], and ambiguously aligned regions were excluded. The resulting FASTA file contains 2417 characters with gaps; ITS contains 482 sites, LSU contains 968 sites, *rpb*2 contains 612 sites, and *tub*2 contains 355 sites. Bayesian inference (BI) and maximum-likelihood (ML) methods were used for phylogenetic analyses of the final FASTA file.

For ML analysis, the built-in file ModelFinder in IQ-TREE was invoked to determine the best model for concatenating genomic FASTA files [62]. TIM2e+I+G4 was used for the final ML search, and maximum-likelihood bootstrap support values (MLBP) were calculated with 1000 replicates.

For BI analysis, the resulting FASTA file was converted to a NEXUS file using MEGA7 [63]. The NEXUS file was executed with MrBayes v. 3.2.2 [64]. The Akaike information criterion (AIC) implemented in jModelTest version 2.0 was used to select the best fit models after likelihood score calculations [65]. HKY+I+G was estimated as the best-fit model under the output strategy of AIC, Lsetnst = 6, rates = gamma. A Markov Chain Monte Carlo (MCMC) algorithm was used to generate phylogenetic trees with Bayesian probabilities. Two runs were executed simultaneously for 6,000,000 generations and sampled every 1000th generation; four chains containing one cold and three heated were run until the average standard deviation of the split frequencies dropped below 0.01, and the stationarity of the analyses was confirmed in line with the standards described by Sun and Guo [66]. The initial 25% of the generations of MCMC sampling were discarded as burn-in. The refinement of the phylogenetic tree was used for estimating BI posterior probability (BIPP) values. FigureTree v1.4.3 was used to visualize the phylogenetic tree [67]. Figure 1 is the final topology of the phylogenetic tree, with BIPP and MLBS of each clade greater than 0.8 and 70%, respectively (BIPP/MLBS).

## 3. Results

### 3.1. Phylogenetic Analysis

All *Didymellaceae* spp. isolates were first identified at the family level based on their ITS sequences. To further identify these isolates at the genus and species level, we conducted four-locus (ITS, LSU, *rpb*2, and *tub*2) phylogenetic analysis referring to 2417 characters and 362 sequences, 310 reference sequences registered from GenBank and 51 representative sequences of *Didymellaceae* isolates obtained in this study. *Coniothyrium palmarum* culture CBS 400.71 served as outgroup. The four individual sequence datasets did not show conflicts in the tree topologies based on preliminary ML analyses, which allowed us to combine the four genes for the multi-locus analysis. The topological structure of the phylogenetic tree constructed by ML and BI was congruent; only the Bayesian tree is shown in this study (Figure 1).

The phylogenetic tree showed that 51 isolates were distributed in 10 genera, including 12 new species, two new varieties and 14 known species. Fifteen isolates fell into the genus *Boeremia*; of these, four isolates clustered with *Boeremia inicola* (BIPP 0.9/MLBS 100), and we designated these isolates as *Boeremia exigua* var. *vulgaris*. Seven isolates were proposed to be new varieties *Boeremia exigua* var. *kasensis* (BIPP 1/MLBS 100); three of them clustered with the known varieties *Boeremia exigua* var. *viburni*, showing 0.99 BIPP and 92% MLBS, and one isolate clustered corresponding to the known species *Boeremia linicola*, showing 1 BIPP and 100% MLBS. Genus *Stagonosporopsis* was enlarged with two new species, *Stagonosporopsis bungeiana* sp. nov., which clustered with *Stagonosporopsis caricae* (BIPP 1/MLBS 100); one isolate clustered with *Stagonosporopsis stuijvenbergii* (BIPP 1/MLBS 100). We designated the isolate as *Stagonosporopsis malaiana* sp. nov. One isolate clustered with the known species *Stagonosporopsis tanaceti*, showing 1 BIPP and 100% MLBS. There was one isolate identified as a known species *Allophoma hayatii* (BIPP 1/MLBS 100) of genus *Allophoma*. Three isolates fell into the genus *Ectophoma*, two isolates were proposed to be a new species, *Ectophoma myriophyllana* sp. nov. (BIPP 0.84/MLBS 99), and one isolate was grouped with the known species *Ectophoma multirostrata* (BIPP 1/MLBS 99). One isolate clustered with *Remotididymella anthropophila* (BIPP 1/MLBS 100), designated as *Remotididymella hydrillana* sp. nov. The strains distributed in the genus *Epicoccum* were identified as four known species: *Epicoccum latusicollum* (BIPP 0.99/MLBS 93), *Epicoccum thailandicum* (BIPP 0.98/MLBS 99), *Epicoccum huancayense* (BIPP 0.94/MLBS 100) and *Epicoccum plurivorum* (BIPP 1/MLBS 100), with only one strain for each. The nineteen isolates that fell into the genus *Didymella* comprised five new species and four known species. Four isolates clustered with *Didymella pomorum* (BIPP 1/MLBS 100); we designated these isolates as *Didymella hippuris* sp. nov.. Nine isolates clustered together near *Didymella dactylidis* and formed three clades representing three new species, which were designated as *Didymella erhaiensis* sp. nov. (BIPP 1/MLBS 100), *Didymella gongkasis* sp. nov. (BIPP 1/MLBS 100), and *Didymella myriophyllana* sp. nov. (BIPP 1/MLBS 100), respectively. Six isolates were distributed among four known species: *Didymella sinensis* (BIPP 1/MLBS 100), *Didymella glomerata* (BIPP 1/MLBS 100), *Didymella degraaffiae* (BIPP 1/MLBS 100) and *Didymella gei* (BIPP 1/MLBS 100). One isolate clustered with *Cumuliphoma indica*, *Cumuliphoma omnivirens* and *Cumuliphoma pneumoniae* (BIPP 1/MLBS 100), and we designated it as *Cumuliphoma lijiangensis* sp. nov. The genus *Leptosphaerulina* was enlarged with two new species, *Leptosphaerulina shangrilensis* sp. nov. (BIPP 1/MLBS 100) and *Leptosphaerulina kasensis* sp. nov. (BIPP 1/MLBS 100), and a known species *Leptosphaerulina saccharicola* (MLBS 99). One isolate clustered with *Dimorphoma saxea* (BIPP 1/MLBS 100), and we designated it as *Dimorphoma isotiana* sp. nov.

### 3.2. Taxonomy

*Boeremia exigua* var. *vulgaris* Y. Huang and Z.F. Yu, sp. nov. (Figure 2).

Etymology: Named after the host species from which the holotype was collected, isolated as an endophyte from the stem of *Hippuris vulgaris* L.

Holotype: China, Sichuan province, Ganzi Tibetan Autonomous Prefecture, Luhuo county, The Kasa Lake Nature Reserve, isolated as an endophyte from the stem of *Hippuris vulgaris* L., September 2015, Y. Huang, Holotype YMF 1.05042.

Sexual morph not observed. Asexual morph on OA. Conidiomata pycnidial, solitary, sometimes aggregated, (sub-)globose, covered with some mycelia, dark olivaceous, superficial or partly immersed in medium, ostiolate, 160–350 × 200–320 μm. Ostiole single, slightly papillate. Pycnidial wall pseudoparenchymatous, composed of oblong to isodiametric cells, 3–6 layers, with outer 2–3 layers pigmented, 18–32 μm thick. Conidiogenous cells phialidic, hyaline, smooth, ampulliform to doliiform, 4–9(–10) × 4–7.5 μm. Conidia oblong to bacilliform, and thin-walled, aseptate, 4.2–7.6 × 2.5–3.5 μm, eguttulate or sometimes with 1–2 guttules per cell. Conidial matrix cream.

Culture characteristics: Colonies on OA, 30–40 mm diam after 7 days, margin regular, covered by dark olivaceous aerial mycelia, densely; reverse concolorous. Colonies on PDA, 30–40 mm diam after 7 days, margin irregular, aerial mycelia dark brown, with a white margin; reverse concolorous. Colonies on MEA, 25–35 mm diam after 7 days, margin irregular, aerial mycelium flat, dark olivaceous, with a white margin; reverse concolorous. Application of NaOH results in a brown discoloration.

Additional specimen examined: China, Sichuan province, Ganzi Tibetan Autonomous Prefecture, Luhuo county, The Kasa Lake Nature Reserve, isolated as an endophyte from the stem of *Hippuris vulgaris*, September 2015, Y. Huang, living culture YMF1.05039; China, Sichuan Province, Ganzi Tibetan Autonomous Prefecture, Litang County, isolated as an endophyte from the leaf of *Hippuris vulgaris*, September 2015, Y. Huang, living culture YMF1.05206; China, Yunnan province, Diqing Tibetan Autonomous Prefecture, Shangrila city, Yila prairie, isolated as an endophyte from the leaf of *Hippuris vulgaris*, September 2015, Y. Huang, living culture YL23.

Notes: Phylogenetically, *Boeremia exigua* var. *vulgaris* was closely related to *Boeremia linicola* Jayaward., Jayasiri and K.D. Hyde. Morphologically, *B. exigua* var. *vulgaris* could be differentiated from *B. linicola* by producing slightly shorter conidia (4.2–7.6 μm vs. 4.90–9.50 μm) [68]. Furthermore, *B. exigua* var. *vulgaris* is distinguishable from *B. linicola* in producing conidia with 1 to 2 polar guttules, while the latter produces conidia with 2 to 4 polar guttules [69].

*Boeremia exigua* var. *kasensis* Y. Huang and Z.F. Yu, sp. nov. (Figure 3).

Etymology: Epithet derived from the location of origin, Kasa Lake in Sichuan province, China.

Holotype: China, Sichuan province, Ganzi Tibetan Autonomous Prefecture, Luhuo county, The Kasa Lake Nature Reserve, isolated as an endophyte from the stem of *Myriophyllum aquaticum* (Vell.) Verdc, September 2016, Y. Huang, Holotype YMF 1.05057.

Sexual morph not observed. Asexual morph on OA. Conidiomata pycnidial, solitary, sometimes aggregated, globose to flask-shaped, covered by some hyphal outgrowths, dark brown to black, superficial or partly immersed in medium, ostiolate, 190–330 × 180–260 μm. Ostiole single, slightly papillate. Pycnidial wall pseudoparenchymatous, composed of oblong to isodiametric cells, 3–6 layers, with outer 2–3 layers pigmented, 18–35 μm thick. Conidiogenous cells phialidic, hyaline, smooth, ampulliform to doliiform, 7.5 × 5 μm. Conidia oblong to bacilliform, or fusiform, aseptate, 4.5–7.5 × 2–3 μm, eguttulate or sometimes with 1–4 guttules per cell. Conidial matrix cream.

Culture characteristics: Colonies on OA, 38–40 mm diam after 7 days, margin irregular, covered by dark brown aerial mycelia, flat, white near the margin; reverse concolorous. Colonies on PDA, 22–28 mm diam after 7 days, margin irregular, aerial mycelia dark brown, white near the margin; reverse brown, white near the margin. Colonies on MEA, 40–42 mm diam after 7 days, margin irregular, brown, white near the margin; reverse brown, white near the margin. Application of NaOH results in a brown to blackish green discoloration.

Additional specimen examined: China, Sichuan province, Ganzi Tibetan Autonomous Prefecture, Luhuo county, The Kasa Lake Nature Reserve, isolated as an endophyte from the stem of *Hippuris vulgaris*, September 2015, Y. Huang, living culture KS18; China, Yunnan province, Diqing Tibetan Autonomous Prefecture, Deqin County, Gongka Lake, isolated as an endophyte from the stem of *Hippuris vulgaris*, September 2015, Y. Huang, living culture YMF1.05031, YMF1.05094, YMF1.05091; China, Sichuan province, Ganzi Tibetan Autonomous Prefecture, Litang County, isolated as an endophyte from the root of *Myriophyllum spicatum* L., September 2016, Y. Huang, living culture YMF1.05205, YMF1.05043.

Notes: Based on multi-locus phylogenetic analysis, seven *Boeremia exigua* var. *kasaensis* isolates formed a solitary clade and near *B. exigua* var. *pseudolilacis* Aveskamp, Gruyter and Verkley. It is distinguished from *B. exigua* var. *pseudolilacis* by the discoloration after application of NaOH to the culture (blackish green discoloration on *B. exigua* var. *kasaensis*, and no effect on *B. exigua* var. *pseudolilacis*), and the conidial matrix is cream instead of rosy-buff [70].

*Stagonosporopsis bungeiana* Y. Huang and Z.F. Yu, sp. nov. (Figure 4).

Etymology: Named after the host species from which the holotype was collected, *Batrachium bungei* (Steud.) L. Liou.

Holotype: China, Yunnan province, Diqing Tibetan Autonomous Prefecture, Shangrila city, Gonka Lake, isolated as an endophyte from the leaf of *Batrachium bungei*, September 2015, Y. Huang. Holotype culture YMF1.05092.

Asexual morph not observed. Sexual morph developed on OA. Ascomata aggregated, sometimes solitary, globose to flask-shaped, brown, small, 200–350 μm diam, papillate, outer wall consisting of 2–5 layers of cells of textura angularis. Pseudoparaphyses not observed. Asci bitunicate, 8-spored, clavate to short cylindrical, 37–62 × 6.5–11 μm. Ascospores biseriate, ellipsoidal, straight to slightly curved, 10.5–14 × 4.5–6 μm, hyaline, smooth, apex obtuse, base broadly obtuse to subobtuse, generally 1-septate, upper cell often wider than lower cell, slightly constricted at the septum.

Culture characteristics: Colonies on OA, 70–75 mm diam after 7 days, margin regular, grayish black, covered by white floccose aerial mycelia; reverse black to grayish white. Colonies on PDA, 80–82 mm diam after 7 days, margin regular, densely covered by white floccose aerial mycelia; reverse white, with concentric circles of grayish white to yellowish. Colonies on MEA, 76–80 mm diam after 7 days, margin regular, white to pale yellow, covered by white floccose aerial mycelia; reverse concolorous. NaOH test negative.

Notes: In the phylogenetic tree, *Stagonosporopsis bungeiana* clustered with *S. caricae* Aveskamp, Gruyter and Verkley in the *Stagonosporopsis* clade. However, *S. bungeiana* could be distinguished from the latter by producing shorter ascospores (10.5–14 μm vs. 17 μm) [71].

*Stagonosporopsis malaiana* Y. Huang and Z.F. Yu, sp. nov. (Figure 5).

Etymology: Named after the host species from which the holotype was collected, *Potamogeton malaianus* Miq.

Holotype: China, Yunnan province, Dali Bai Autonomous Prefecture, Dali city, Erhai Lake, isolated as an endophyte from the stem of *Potamogeton malaianus* Miq., September 2015, Y. Huang. Holotype culture YMF1.05087.

Sexual morph not observed. Asexual morph on OA. Conidiomata pycnidial, mostly aggregated, some solitary, flask-shaped or sub-globose, covered by some hyphal outgrowths, dark brown, superficial or partly immersed in medium, ostiolate, 150–560 × 140–520 μm. Ostiole single, slightly papillate, sometimes elongated as a short neck. Pycnidial wall pseudoparenchymatous, composed of oblong to isodiametric cells, 3–6 layers, with outer 2–3 layers pigmented, 25–50 μm thick. Conidiogenous cells phialidic, hyaline, smooth, subglobose, ampulliform or lageniform, 5.5–7 × 4–6 μm. Conidia ovoid, oblong with a rounded or cylindrical end, aseptate, 3.2–4.5 × 1.5–2.4 μm, mostly with 1–2 guttules per cell. Conidial matrix cream.

Culture characteristics: Colonies on OA, 68–70 mm diam after 7 days, margin regular, covered by greenish olivaceous aerial mycelia, dense, with concentric circles of pale olivaceous, smoke gray near the center; reverse olivaceous to white. Colonies on PDA, 65–69 mm diam after 7 days, margin regular, aerial mycelia woolly, dark gray to white, forming some radial lines near the center; reverse gray to white. Colonies on MEA, 59–68 mm diam after 7 days, margin regular, olivaceous, with sparse white aerial mycelia near the center; reverse concolorous. Application of NaOH results in a reddish brown discoloration of the agar.

Notes: *Stagonosporopsis malaiana* is phylogenetically closely related to *S. stuijvenbergii* Hern.-Restr., L.W. Hou, L. Cai and Crous, but differs morphologically from the latter by producing shorter conidia (3.2–4.5 × 1.5–2.4 μm vs. 3.5–6.5 × 2–3 µm) [35].

*Didymella hippuris* Y. Huang and Z.F. Yu, sp. nov. (Figure 6).

Etymology: Named after the host genus *Hippuris*, from which the holotype was isolated.

Holotype: China, Yunnan province, Diqing Tibetan Autonomous Prefecture, Deqin County, Gongka Lake, isolated as an endophyte from the stem of *Hippuris vulgaris*, September 2015, Y. Huang. Holotype culture YMF1.05089.

Sexual morph not observed. Asexual morph on OA. Conidiomata pycnidial, solitary, sometimes aggregated, globose to flask-shaped, brown, with some white hyphal outgrowths, superficial or partly immersed in medium, ostiolate, 80–420×120–490 μm. Ostiole single, non-papillate or slightly papillate. Pycnidial wall pseudoparenchymatous, composed of oblong to isodiametric cells, 3–6 layers, with outer 2–3 layers pigmented, 18–32 μm thick. Chlamydospores dark brown, 5.4–10 × 3.8–7.7 µm. Conidiogenous cells phialidic, hyaline, smooth, ampulliform to doliiform, 4.5–13.5 × 3.5–8.5 μm. Conidia oblong to bacilliform, or fusiform, aseptate, 5–9 × 2.5–3.6 μm, eguttulate or sometimes with 1–2 guttules per cell. Conidial matrix brown.

Culture characteristics: Colonies on OA, 64–68 mm diam after 7 days, margin regular, dark brown, covered by floccose aerial mycelia, whitish; reverse brown. Colonies on PDA, 69–70 mm diam after 7 days, margin regular, aerial mycelia woolly, grayish white; reverse brown, white near the margin. Colonies on MEA, 65–67 mm diam after 7 days, margin regular, covered by floccose aerial mycelia, dense, pale brown to white; reverse dark brown to white. NaOH test negative.

Additional specimen examined: China, Yunnan province, Diqing Tibetan Autonomous Prefecture, Shangrila city, September 2015, Y. Huang, living culture YMF1.05210; China, Sichuan province, Ganzi Tibetan Autonomous Prefecture, Luhuo county, The Kasa Lake Nature Reserve, isolated as an endophyte from the stem of *Myriophyllum spicatum*, September 2016, Y. Huang, living culture YMF1.05204; China, Sichuan province, Ganzi Tibetan Autonomous Prefecture, Luhuo county, The Kasa Lake Nature Reserve, isolated as an endophyte from the root of *Hippuris vulgaris*, September 2016, Y. Huang, living culture YMF1.05037.

Notes: Based on multi-locus phylogenetic analysis, four strains of *Didymella hippuris* formed a solitary clade, and phylogenetically near *Did. Pomorum* Qian Chen and L. Cai, but it can be distinguished by the conidial matrix being brown rather than cream white [72]. Furthermore, the conidia of *Did. hippuris* are wider than those of *Did. pomorum* (2.5–3.6 μm vs. 1.5–2.5(–3) µm) [73].

*Didymella erhaiensis* Y. Huang and Z.F. Yu, sp. nov. (Figure 7).

Etymology: Epithet derived from the location of origin, Erhai Lake, Yunnan province, China.

Holotype: China, Yunnan province, Dali Bai Autonomous Prefecture, Dali city, Erhai Lake, isolated as an endophyte from the leaf of *Eichhornia crassipes* (Mart.) Solms, September 2015, Y. Huang. Holotype culture YMF1.05021.

Sexual morph not observed. Asexual morph on OA. Conidiomata pycnidial, solitary, sometimes aggregated, globose to (sub-)globose, grayish brown, superficial or partly immersed in medium, ostiolate, 90–400 × 120–380 μm. Ostiole 1–2, slightly papillate, sometimes elongated as a short neck. Pycnidial wall pseudoparenchymatous, composed of oblong to isodiametric cells, 3–6 layers, with outer 2–3 layers pigmented, 20–45 μm thick. Conidiogenous cells phialidic, hyaline, smooth, ampulliform to doliiform, 6–7 × 5–6 μm. Conidia oblong to bacilliform, or fusiform, aseptate, 3.5–5 × 1.8–2.5 μm, eguttulate or sometimes with 1–3 guttules per cell. Conidial matrix brown.

Culture characteristics: Colonies on OA, 60–67 mm diam after 7 days, margin regular, grayish brown, covered by flat and gray aerial mycelia; reverse brown. Colonies on PDA, 60–70 mm diam after 7 days, margin regular, grayish brown; reverse dark brown. Colonies on MEA, 58–66 mm diam after 7 days, margin irregular, dark brown, covered by brown aerial mycelia, some radially furrowed zones near the center; reverse dark brown, white near the margin. Application of NaOH results in a brown discoloration of the agar.

Additional specimen examined: China, Yunnan province, Dali Bai Autonomous Prefecture, Dali city, Erhai Lake, isolated as an endophyte from the root of *Sonchus oleraceus* L., September 2015, Y. Huang, living culture YMF1.05084; China, Yunnan province, Dali Bai Autonomous Prefecture, Dali city, Erhai Lake, isolated as an endophyte from the root of *Hydrocharis dubia* (Bl.) Backer, September 2015, Y. Huang, living culture YMF1.05023; China, Yunnan province, Dali Bai Autonomous Prefecture, Dali city, Erhai Lake, isolated as an endophyte from the leaf of *Eichhornia crassipes*, September 2015, Y. Huang, living culture YMF1.05024.

Notes: Phylogenetically, four strains of *Didymella erhaiensis* formed a solitary clade, and this species was closely related to *Did. gongkasis* and *Did. dactylidis*. Morphologically, *Did*. *erhaiensis* is different from these two species by producing slightly shorter conidia (3.5–5 μm vs. 4.5–6 μm vs. 4.5–9(–9) μm). Moreover, *Did. erhaiensis* is different from *Did. dactylidis* by producing conidia with 1–3 guttules instead of (2–)4–8(–15) guttules [27].

*Didymella myriophyllana* Y. Huang and Z.F. Yu, sp. nov. (Figure 8).

Etymology: Named after the host species from which the holotype was collected, isolated as an endophyte from the stem of *Myriophyllum aquaticum.*

Holotype: China, Sichuan province, Ganzi Tibetan Autonomous Prefecture, Luhuo county, The Kasa Lake Nature Reserve, isolated as an endophyte from the leaf of *Myriophyllum aquaticum*, September 2016, Y. Huang. Holotype culture YMF1.05100.

Sexual morph not observed. Asexual morph on OA. Conidiomata pycnidial, solitary, sometimes aggregated, scattered, globose to flask-shaped, pale brown at first, then turned dark brown, superficial or partly immersed in medium, ostiolate, 200–430 × 180–430 μm. Ostiole single, slightly papillate, sometimes elongated as a short neck. Pycnidial wall pseudoparenchymatous, composed of oblong to isodiametric cells, 3–6 layers, with outer 2–3 layers slightly pigmented, 25–55 μm thick. Conidiogenous cells phialidic, hyaline, smooth, ampulliform to doliiform, 6–7 × 5–6 μm. Conidia oblong to bacilliform, or fusiform, aseptate, 4–6 × 1.6–2.5 μm, eguttulate or sometimes with 1–3 guttules per cell. Conidial matrix brown.

Culture characteristics: Colonies on OA, 50–55 mm diam after 7 days, margin regular, covered by flat aerial mycelia, brown to pale yellow; reverse brown, pale yellow near the margin. Colonies on PDA, 45–50 mm diam after 7 days, margin irregular, grayish brown, covered by floccose aerial mycelia; reverse black, grayish white near the margin. Colonies on MEA, 40–45 mm diam after 7 days, margin irregular, covered by floccose aerial mycelia, blackish green to dark brown, pale yellow near the margin; reverse black, white near the margin. NaOH test negative.

Additional specimen examined: China, Sichuan province, Ganzi Tibetan Autonomous Prefecture, isolated as an endophyte from the leaf of *Hippuris vulgaris*, September 2016, Y. Huang, living culture GZ58; China, Sichuan province, Ganzi Tibetan Autonomous Prefecture, Luhuo county, The Kasa Lake Nature Reserve, isolated as an endophyte from the leaf of *Myriophyllum aquaticum*, September 2016, Y. Huang. living culture YMF1.05035.

Note: Based on multi-locus phylogenetic analysis, two strains of *Didymella myriophyllana* formed a solitary clade, and near a new species *Did. gongkasis*. Morphologically, the conidia of *Did. myriophyllana* are narrower than those of *Did. gongkasis*, 1.6–2.5 μm vs. 2–2.8 µm. Moreover, this species is distinguishable from *Did. gongkasis* by the absence of a positive reaction to NaOH.

*Leptosphaerulina shangrilensis* Y. Huang and Z.F. Yu, sp. nov. (Figure 9).

Etymology: Epithet derived from the location of origin, Diqing Tibetan Autonomous Prefecture, Shangrila city in Yunnan province, China.

Holotype: China, Yunnan province, Diqing Tibetan Autonomous Prefecture, Shangrila city, isolated as an endophyte from the root of *Hippuris vulgaris*, September 2015, Y. Huang. Holotype culture YMF1.05053.

Sexual morph developed on OA. Ascomata solitary or aggregated, brown, superficial or partly immersed in medium, subglobose or obpyriform, 80–240 × 80–230 μm. Ostiole near the center, 30 μm diam. Asci bitunicate, 8-spored, hyaline, obovoid, 42–90 × 35–50 μm. Ascospores multi-seriate, ellipsoidal to obovoid, mucilaginous sheaths, 27–41 × 12–18 μm, hyaline, smooth, apex obtuse, base broadly obtuse to subobtuse, 5 transverse septate, 2–3 longitudinal septate, upper cell often wider than lower cell, slightly constricted at the septum. Hamathecium lacking pseudoparaphyses.

Culture characteristics: Colonies on OA, 33–35 mm diam after 7 days, margin regular, grayish black, covered by grayish white aerial mycelia, sparse; reverse black. Colonies on PDA, 38–45 mm diam after 7 days, margin regular, dark brown with some gray section, covered by floccose aerial mycelia; reverse black, white near the margin. Colonies on MEA, 40–41 mm diam after 7 days, margin regular, dark brown with some grayish white section, covered by floccose aerial mycelia; reverse black. Application of NaOH results in a brown to reddish brown discoloration.

Note: *Leptosphaerulina shangrilensis* is closely related to *L. arachidicola* W.Y. Yen, M.J. Chen and K.T. Huang in phylogeny. Morphologically, *L. shangrilensis* is distinguished from *L. arachidicola* by septation of ascospores. For instance, *L. shangrilensis* regularly produces ascospores up to 5 transverse septate and 2–3 longitudinal septate; whereas *L. arachidicola* produces ascospores up to 3–5 transverse septate and 0–2 longitudinal septate [74,75].

*Leptosphaerulina kasensis* Y. Huang and Z.F. Yu, sp. nov. (Figure 10).

Etymology: Epithet derived from the location of origin, Kasa Lake in Sichuan province, China.

Holotype: China, Sichuan province, Ganzi Tibetan Autonomous Prefecture, Luhuo county, The Kasa Lake Nature Reserve, isolated as an endophyte from the leaf of *Hippuris vulgaris*, September 2015, Y. Huang. Holotype culture YMF1.05041.

Sexual morph developed on OA. Ascomata solitary or aggregated, brown, superficial or partly immersed in medium, subglobose or obpyriform, 120–210 × 100–160 μm. Ostiole near the center, 30 μm diam. Asci bitunicate, 8-spored, hyaline, obovoid, 90–120 × 40–50 μm. Ascospores multi-seriate, ellipsoidal to obovoid, mucilaginous sheaths, 28–33 × 13–15 μm, hyaline, smooth, apex obtuse, base broadly obtuse to subobtuse, 4 transverse septate, 2–3 longitudinal septate, upper cell often wider than lower cell, slightly constricted at the septum. Hamathecium lacking pseudoparaphyses.

Culture characteristics: Colonies on OA, 55–60 mm diam after 7 days, margin irregular, grayish black, covered by grayish white aerial mycelia, sparse; reverse black. Colonies on PDA, 35–45 mm diam after 7 days, margin irregular, brown, covered by gray aerial mycelia; reverse brown. Colonies on MEA, 38–45 mm diam after 7 days, margin regular, dark brown, covered by grayish white aerial mycelia; reverse brown. Application of NaOH results in a brown to reddish brown discoloration.

Notes: In the phylogenetic tree, *Leptosphaerulina kasensis* formed a distinct lineage, closely related to *L. australis* McAlpine, but can be easily distinguished from the latter by producing narrower ascomata (100–160 μm vs. 160–180 μm) and narrower ascospores (13–15 μm vs. 30–43 μm) [76]. *L. kasensis* resembles *L. gaeumannii* Cec. Roux, Trans Br, *L. macrospora* J.M. Liang and L. Cai, and *L. saccharicola* Phookamsak., J.K. Liu and K.D. Hyde in having ascospores with similar size. However, these species can be distinguished by having ascospores in different numbers of longitudinal septa and transverse septa [59,74]. *L. kasensis* is also easily distinguished from *L. longiflori* Tennakoon, C.H. Kuo and K.D Hyde by the size of the ascospores (28–33 × 13–15 μm vs. (25–)27–32(–35.5) × (9–)10–11.5(–12) µm) [77].

*Cumuliphoma lijiangensis* Y. Huang and Z.F. Yu, sp. nov. (Figure 11).

Etymology: Epithet derived from the location of origin, Lijiang City in Yunnan province, China.

Holotype: China, Yunnan province, Lijiang city, isolated as an endophyte from the leaf of *Myriophyllum spicatum*, September 2016, Y. Huang. Holotype culture YMF1.05096.

Sexual morph not observed. Asexual morph on OA. Conidiomata pycnidial, solitary, sometimes aggregated, globose to flask-shaped, superficial or partly immersed in medium, ostiolate, 80–300 × 100–250 μm. Ostiole single, slightly papillate, sometimes elongated as a short neck. Pycnidial wall pseudoparenchymatous, composed of oblong to isodiametric cells, 3–6 layers, with outer 2–3 layers pigmented, 25–50 μm thick. Conidiogenous cells phialidic, hyaline, smooth, ampulliform to doliiform, 6–7 × 5–6 μm. Conidia oblong to bacilliform, or fusiform, aseptate, 3–4 × 1.5–2.2 μm, eguttulate or sometimes with 1–2 guttules per cell. Conidial matrix brown.

Culture characteristics: Colonies on OA, 50–56 mm diam after 7 days, margin regular, brown with some grayish white section, covered by floccose aerial mycelia; reverse dark brown to pale brown. Colonies on PDA, 53–55 mm diam after 7 days, margin regular, pale brown with some grayish white section, covered by floccose aerial mycelia; reverse brown, grayish white near the margin. Colonies on MEA, 40–42 mm diam after 7 days, margin regular, brown, covered by floccose aerial mycelia, white near the margin; reverse concolorous. Application of NaOH results in a blackish green to dark brown discoloration.

Note: In the phylogenetic tree *Cumuliphoma lijiangensis*, which was isolated from Lijiang city, formed an independent lineage basal to the genus *Cumuliphoma*, and *C. lijiangensis* as a sister taxon to *C. indica* Valenz.-Lopez, Stchigel, Crous, Guarro and Cano, *C. omnivirens* Valenz.-Lopez, Stchigel, Crous, Guarro and Cano and *C. pneumoniae* Valenz.-Lopez, Stchigel, Crous, Guarro and Cano with strong statistical support (98% ML, 0.98 BIPP) (Figure 1). This species is morphologically closely related to *C. omnivirens*. However, *C. lijiangensis* does not produce chlamydospores [78]. Also, ITS sequence comparison revealed 3 base pair differences between *C. lijiangensis* and *C. indica*, and 4 base pair differences between *C. lijiangensis* and *C. pneumoniae* [78,79].

*Dimorphoma isotiana* Y. Huang and Z.F. Yu, sp. nov. (Figure 12).

Etymology: Named after the host genus *Isotes*, from which the holotype was isolated.

Holotype: China, Yunnan province, Diqing Tibetan Autonomous Prefecture, Shangrila city, isolated as an endophyte from the root of *Isoetaceae*, September 2015, Y. Huang. Holotype culture YMF 1.05048.

Sexual morph not observed. Asexual morph on OA. Conidiomata pycnidial, solitary, sometimes aggregated, globose to flask-shaped, superficial or partly immersed in medium, ostiolate, 140–300 × 140–330 μm. Ostiole single, slightly papillate, sometimes elongated as a short neck. Pycnidial wall pseudoparenchymatous, composed of oblong to isodiametric cells, 3–6 layers, with outer 2–3 layers pigmented, 18–35 μm thick. Conidiogenous cells phialidic, hyaline, smooth, ampulliform to doliiform, 4–6 × 3.5–5 μm. Conidia globose, cylindrical to ellipsoidal, aseptate, 4–6 × 2.5–4 μm, eguttulate or sometimes with 1–2 guttules per cell. Conidial matrix cream.

Culture characteristics: Colonies on OA, 55–60 mm diam after 7 days, margin regular, covered by flat aerial mycelia, dark olivaceous, gray near the center; reverse concolorous. Colonies on PDA, 50–55 mm diam after 7 days, margin irregular, aerial mycelia greenish olivaceous, white near the margin; reverse concentric circles of different color, dark olivaceous near the center, dark brown to pale yellow. Colonies on MEA, 55–60 mm diam after 7 days, margin regular, yellowish brown, white near the center; reverse brown, white near the margin. Application of NaOH results in a brown to reddish discoloration.

Note: Phylogenetically, *Dimorphoma isotiana* formed a distinct lineage near *Dim. saxea* L.W. Hou, L. Cai and Crous. Morphologically, *Dimorphoma* formed an independent lineage clearly distinct from *Xenodidymella* and other genera in *Didymellaceae* by producing an extremely thin pycnidial wall and globose, cylindrical to ellipsoidal conidia [70]. Furthermore, *Dim*. *isotiana* was characterized by having pycnidial wall 18–35 μm thick, while *Dim. saxea* had pycnidial wall 10–17 μm thick [27].

*Didymella gongkaensis* Y. Huang and Z.F. Yu, sp. nov. (Figure 13).

Etymology: Epithet derived from the location of origin, Gongka Lake Yunnan, China.

Holotype: China, Yunnan province, Diqing Tibetan Autonomous Prefecture, Deqin County, Gongka Lake, isolated as an endophyte from the leaf of *Hippuris vulgaris*, September 2015, Y. Huang. Holotype culture YMF1.05029.

Sexual morph not observed. Asexual morph on OA. Conidiomata pycnidial, solitary, sometimes aggregated, globose to flask-shaped, superficial or partly immersed in medium, ostiolate, 200–250 × 110–130 μm. Ostiole single, slightly papillate, sometimes elongated as a short neck. Pycnidial wall pseudoparenchymatous, composed of oblong to isodiametric cells, 3–6 layers, with outer 2–3 layers pigmented, 25–50 μm thick. Conidiogenous cells phialidic, hyaline, smooth, ampulliform to doliiform, 6–7 × 5–6 μm. Conidia oblong to bacilliform, or fusiform, aseptate, 4.5–6 × 2–2.8 μm, eguttulate or sometimes with 1–3 guttules per cell. Conidial matrix brown.

Culture characteristics: Colonies on OA, 50–60 mm diam after 7 days, margin regular, grayish brown, covered by floccose aerial mycelia, pale yellow near the center; reverse brown. Colonies on PDA, 52–60 mm diam after 7 days, margin regular, covered by floccose aerial mycelia, grayish brown; reverse dark brown. Colonies on MEA, 45–50 mm diam after 7 days, margin regular, dark brown, covered by floccose aerial mycelia, white near the margin; reverse concolorous. Application of NaOH results in a brown to reddish discoloration.

Additional specimen examined: China, Yunnan province, Diqing Tibetan Autonomous Prefecture, Deqin County, Gongka Lake, isolated as an endophyte from the leaf of *Hippuris vulgaris*, September 2015, Y. Huang, living culture YMF1.05095.

Note: Phylogenetically, two strains of *Didymella gongkaensis* formed a distinct lineage, and this species was closely related to new species *Did. myriophyllana*. Morphologically, *Did. gongkaensis* could be differentiated from *Did. myriophyllana* by producing slightly larger conidia (4.5–6 × 2–2.8 μm vs. 4–6 × 1.6–2.5 μm).

*Ectophoma myriophyllana* Y. Huang and Z.F. Yu, sp. nov. (Figure 14).

Etymology: Named after the host species from which the holotype was collected, *Myriophyllum spicatum*.

Holotype: China, Sichuan province, Liangshan Yi Autonomous Prefecture, Xichang city, Qionghai, isolated as an endophyte from the leaf of *Myriophyllum spicatum*, September 2015, Y. Huang. Holotype culture YMF1.05050.

Sexual morph not observed. Asexual morph on OA. Conidiomata pycnidial, solitary, sometimes aggregated, globose to flask-shaped, covered by some hyphal outgrowths, superficial or partly immersed in medium, ostiolate, 160–350 × 200–320 μm. Ostiole single, slightly papillate. Pycnidial wall pseudoparenchymatous, composed of oblong to isodiametric cells, 3–6 layers, with outer 2–3 layers pigmented, 20–30 μm thick. Conidiogenous cells phialidic, hyaline, smooth, doliiform, 3.5–7 × 3.3–4.5 μm. Conidia oblong to bacilliform, 3–4 × 1.5–2 μm, with 1–2 guttules per cell. Conidial matrix dark brown.

Culture characteristics: Colonies on OA, 68–70 mm diam after 7 days, margin regular, dark brown near the center, with a greenish olivaceous margin, forming concentric rings; reverse concolorous. Colonies on PDA, 65–72 mm diam after 7 days, margin regular, covered by floccose aerial mycelia, pale brown to gray, forming concentric rings; reverse concolorous. Colonies on MEA, 68–71 mm diam after 7 days, margin regular, covered by floccose aerial mycelia, dark brown to gray, forming concentric rings; reverse dark brown. Application of NaOH results in a brown discoloration.

Additional specimen examined: China, Sichuan province, Liangshan Yi Autonomous Prefecture, Xichang city, Qionghai, isolated as an endophyte from the leaf of *Elodea nuttallii* (Planch.) H.St.John, September 2015, Y. Huang, living culture YMF1.05208

Note: Two strains of *Ectophoma myriophyllana* clustered a solitary clade, isolated from different plant collected from Qionghai, and phylogenetically near *E. multirostrata* Valenz.-Lopez, Cano, Crous, Guarro and Stchigel, but it can be easily distinguished from them in morphology. For instance, *E. myriophyllana* could be differentiated from *E. multirostrata* by producing narrower conidia (1.5–2 μm vs. 4.5–4.2 μm) [80].

*Remotididymella hydrillana* Y. Huang and Z.F. Yu, sp. nov. (Figure 15).

Etymology: Epithet after the host species from which the holotype was collected, *Hydrilla verticillata* (Linn. f.) Royle.

Holotype: China, Yunnan Province, Dali Bai Autonomous Prefecture, Dali City, Erhai, isolated as an endophyte from the stem of *Hydrilla verticillata*, September 2015, Y. Huang. Holotype culture YMF1.05022.

Sexual morph not observed. Asexual morph on OA. Conidiomata pycnidial, solitary, sometimes aggregated, flask-shaped, covered by some hyphal outgrowths, superficial or partly immersed in medium, ostiolate,180–350 × 140–300 μm. Ostiole single, slightly papillate. Pycnidial wall pseudoparenchymatous, composed of oblong to isodiametric cells, 3–6 layers, with outer 2–3 layers pigmented, 25–35 μm thick. Conidiogenous cells phialidic, hyaline, smooth, ampulliform to doliiform, 5.5–9 × 4.5–5.5 μm. Conidia oblong to bacilliform, or fusiform, aseptate, 4–6 × 1.5–2 μm, with 1–2 guttules per cell. Conidial matrix brown.

Culture characteristics: Colonies on OA, 65–70 mm diam after 7 days, margin regular, pale brown with some black section, covered by floccose aerial mycelia; reverse dark brown. Colonies on PDA, 65–68 mm diam after 7 days, margin regular, aerial mycelia floccose, white; reverse brown, with a grayish white margin. Colonies on MEA, 60–70 mm diam after 7 days, margin regular, pale brown, covered by floccose aerial mycelia; reverse white to pale yellow, forming concentric rings. NaOH test negative.

Note: Phylogenetically, *Remotididymella hydrillana* formed a distinct lineage near *R. anthropophila* Valenz.-Lopez, Cano, Guarro and Stchigel. Morphologically, *R. hydrillana* differs from *R. anthropophila* in producing longer conidia (5.5–7.5 μm vs. 4–6 μm) [78].

## 4. Discussion

In recent years, the vast majority of studies on the diversity of *Didymellaceae* have focused on the soil environment [23,25,36,78,81,82]. In contrast, the diversity of this family in freshwater ecosystems has received very little attention. In our study, we investigated the species diversity of the *Didymellaceae* fungi of aquatic plants in southwestern China, and the sampling sites included wetlands, lakes and ponds in Yunnan, Sichuan and Guizhou provinces. Among 51 *Didymellaceae* strains, 33 isolates represented 12 new species and two new varieties, and high frequency occurrence of new species showed that aquatic plants may be a special ecological niche that highly promotes species differentiation. New species were distributed in 10 genera, mainly in *Boeremia*, *Didymella, Epicoccum*, *Leptosphaerulina* and *Stagonosporopsis*, and these dominant genera are also frequently reported in terrestrial ecosystems [23,70].

*Didymella* was the most abundant genus in the study and was isolated from seven aquatic plants collected from 12 sampling sites. Previously, *Didymella* was reported as a worldwide fungus that often takes advantage of specific conditions to colonize plants, occasionally causing serious damage, such as stem cankers and black spot disease of fruits and leaves [83,84]. At the same time, *Didymella* is the only saprophytic genus that is related to *Ascochyta* and *Phoma* [22,27,85], and the question of the polyphyly of *Ascochyta*, *Didymella* and *Phoma* remains unresolved [23]. In addition, *Boeremia* was another abundant genus, with 15 strains belonging to *Boeremia.* Species of *Boeremia* are morphologically similar to *Phoma*, and belonged to the *Phoma* genus previously. Species of *Boeremia* have been isolated from plants as pathogens or endophytes with a worldwide occurrence, mainly associated with rots of various organs [86]. One of the most well-known varieties is *B. exigua* var. *exigua*, which is reported to be a pathogen of more than 200 plant species, causing leaf blight, leaf spot, and connection to post-harvest diseases [87]. The genus was also reported to be isolated from sweet potatoes with leaf spot symptoms in China and Brazil [88,89]. In this study, two new varieties were described, namely *B. exigua* var. *vulgaris* and *B. exigua* var. *kasensis.*

*Didymellaceae* is an important group of fungi with a wide host distribution, and legumes, grasses, *Asteraceae*, buttercups, *Rosaceae*, and *Solanaceae* are the most common hosts [22,24,25,27,90]. Chen et al. [25] carried out a correlation analysis between *Didymellaceae* and host plants. The results indicated that only a few genera showed some degree of host specificity, e.g., *Ascochyta* exclusively infected legumes, and *Gramineae* and *Cruciferae* were host-specific, with *Neoascochyta* and *Phomatodes* as hosts, respectively. In the current study, the strains of *Didymellaeae* had no obvious host specificity. Fifty-one strains were isolated from 12 host genera of nine families, and plants of *Haloragidaceae*, *Hippuridaceae*, *Hydrocharitaceae*, and *Potamogetonaceae* were common hosts (Appendix A). At the species level, *Hippuris vulgaris* was the most common host, with 25 isolates belonging to 14 species of four genera inhabiting this plant, showing that endophytes from aquatic plants had no host specificity in our study. In terms of the frequency of fungi isolated from the three plant tissues (roots, stems, and leaves), fungi were isolated more frequently from stems and leaves than from roots (Appendix A), possibly because roots inhabit an anoxic environment compared to the living environment of stems and leaves.

In our isolates, we found three new species, *S. bungeiana*, *L. shangrilensis*, and *L. kasensis*, characterized by their teleomorphic stage. The *Leptosphaerulina* species usually have small, dark pseudothecia with membraneous walls and hyaline, and dictyosporous ascospores [59]. Excluding *L. saccharicola* and *L. australis*, *Leptosphaerulina* species are often observed in the teleomorphic stages [74,91]. *Stagonosporopsis* is mainly regarded as a causal agent in different plants, such as gummy stem blight in pumpkins and ray blight in pyrethrum [92,93]. Typical characteristics of *Stagonosporopsis* species are their oblong to ellipsoid conidia, as well as the sub-globose conidiomata with slight papillate [82]. In addition, *S. malaiana* was identified and introduced as a new species belonging to this genus.

In the last decade, our knowledge of the *Didymellaceae* fungi and their relationships with plant hosts has grown exponentially due to advances in bioinformatics and molecular phylogenetics. As a family of phytopathological importance, they have an important impact on agricultural production. Therefore, it is important to reveal a large number of species and to enrich the species resources in this family. In the present study, a large-scale survey of the *Didymellaceae* fungi was carried out, demonstrating the presence of a large number of unknown species in some previously neglected ecosystems, and illustrating that the diversity of this family in aquatic plants is much higher than currently estimated. Despite the success of combining four loci (LSU, ITS, *rpb*2 and *tub*2), to analyze the phylogenetic relationships of *Didymellaceae*, there are still some classification statuses to be further determined, such as complex species in *Didymella* and *Boeremia exigua* varieties. More genetic loci and isolates are needed to elucidate in detail their phylogenetic relationships and their species boundaries.

## Figures and Tables

**Figure 1 jof-09-00761-f001:**
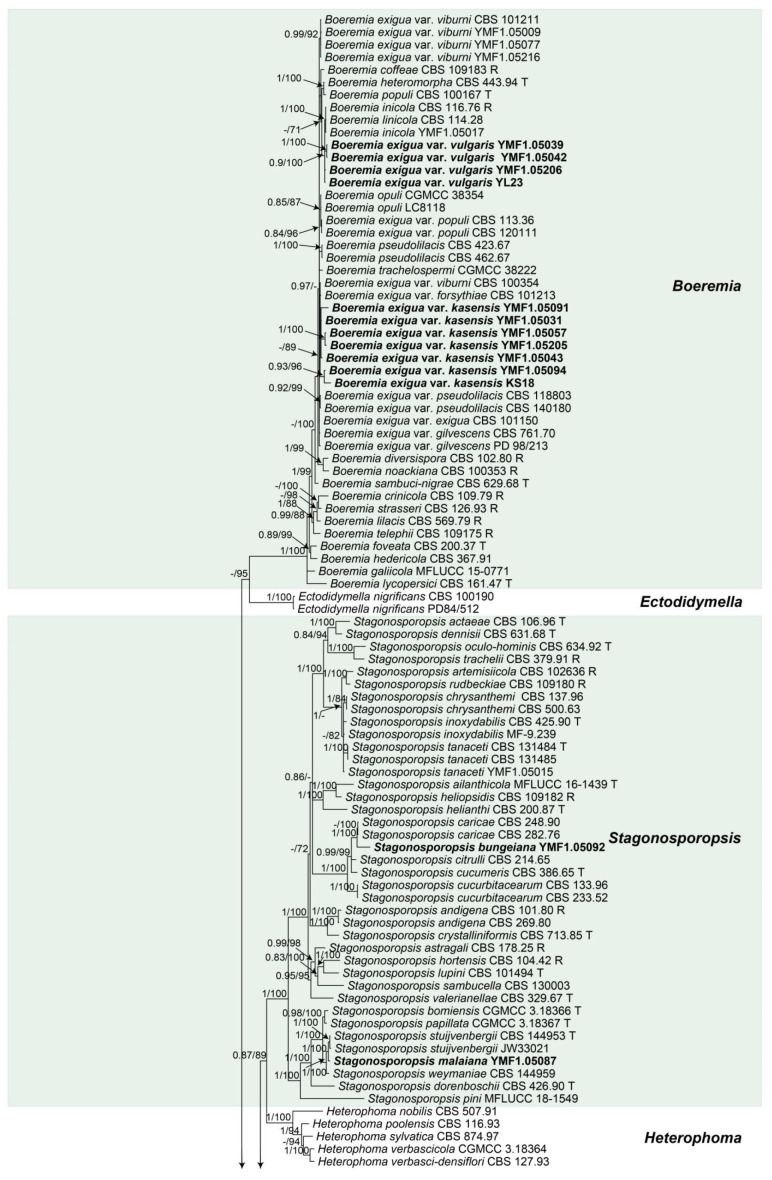
Phylogenetic tree inferred from a Bayesian analysis based on a concatenated alignment of ITS, LSU, *rpb*2, and *tub*2 sequences of 362 strains representing *Didymellaceae* and outgroup sequence. The Bayesian posterior probabilities (BIPP) above 0.80 and RAxML bootstrap support values (MLBS) above 70% are given at the nodes (BIPP/MLBS). Some of the basal branches were shortened to facilitate layout. Genera are delimited in colored boxes, with the genus name indicated to the right. Strains with special status are indicated with a superscript letter after the accession number (R: representative; T: ex-type). The new species obtained in this research are printed in bold. The tree is rooted to *Coniothyrium palmarum* culture CBS 400.71.

**Figure 2 jof-09-00761-f002:**
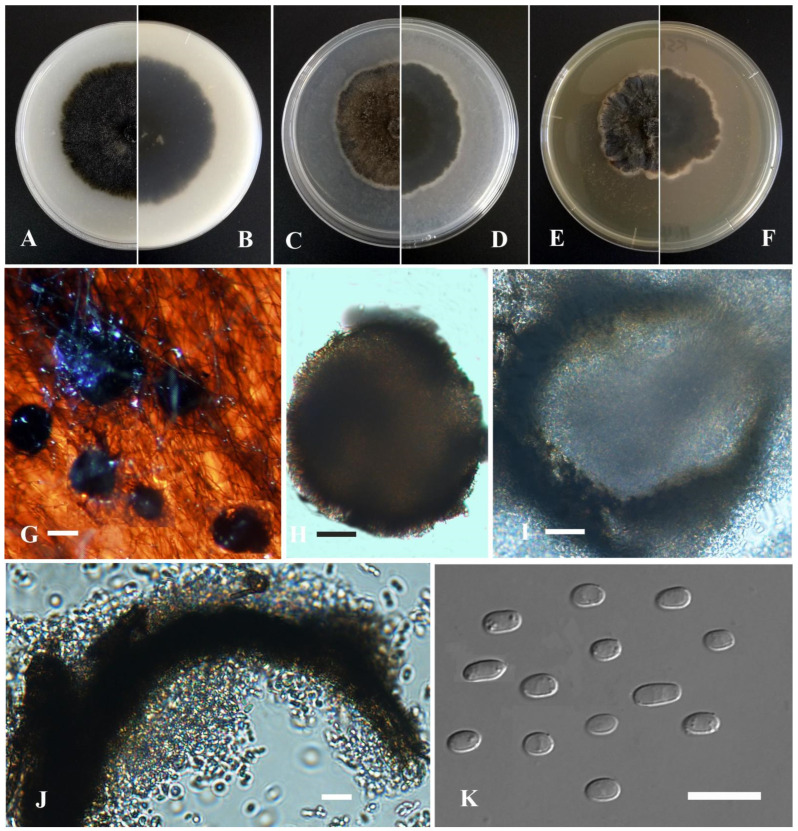
*Boeremia exigua* var. *vulgaris* (YMF1.05042). (**A**,**B**) Colony on OA medium (front and reverse) after 7 days. (**C**,**D**) Colony on PDA medium (front and reverse) after 7 days. (**E**,**F**) Colony on MEA medium (front and reverse) after 7 days. (**G**) Pycnidia on OA medium. (**H**) Pycnidia. (**I**) Section of pycnidia. (**J**) Pycnidial wall. (**K**) Conidia. Scale bar: (**G**) = 100 μm; (**H**,**I**) = 50 μm; (**J**) = 20 μm; (**K**) = 10 μm.

**Figure 3 jof-09-00761-f003:**
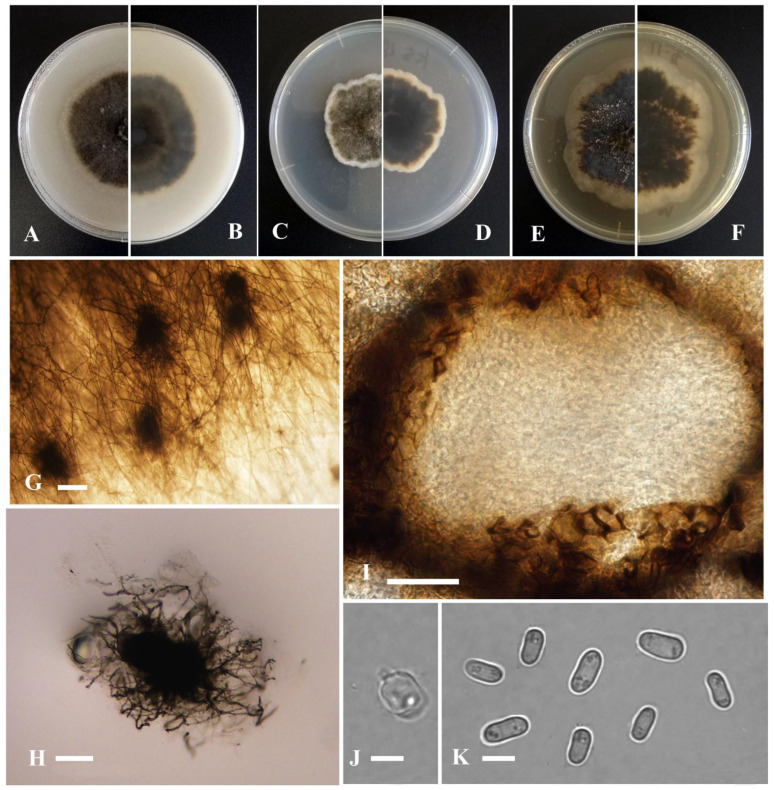
*Boeremia exigua* var. *kasensis* (YMF1.05057). (**A**,**B**) Colony on OA medium (front and reverse) after 7 days. (**C**,**D**) Colony on PDA medium (front and reverse) after 7 days. (**E**,**F**) Colony on MEA medium (front and reverse) after 7 days. (**G**) Pycnidia on OA medium. (**H**) Pycnidia. (**I**) Section of pycnidia. (**J**) Conidiogenous cell. (**K**) Conidia. Scale bar: (**G**) = 100 μm; (**H**,**I**) = 50 μm; (**J**,**K**) = 5 μm.

**Figure 4 jof-09-00761-f004:**
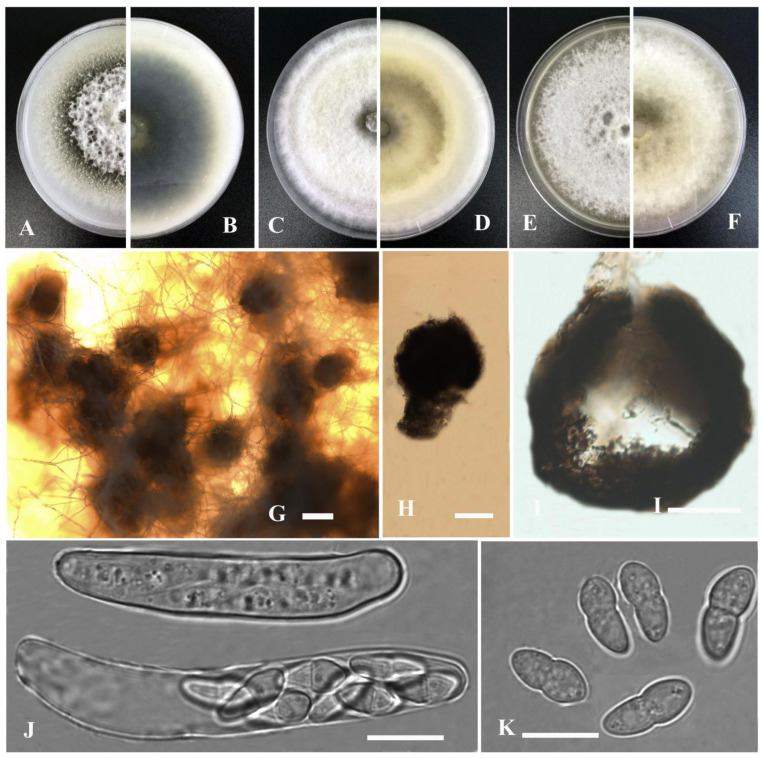
*Stagonosporopsis bungeiana* (YMF1.05092). (**A**,**B**) Colony on OA medium (front and reverse) after 7 days. (**C**,**D**) Colony on PDA medium (front and reverse) after 7 days. (**E**,**F**) Colony on MEA medium (front and reverse) after 7 days. (**G**) Ascomata on OA medium. (**H**) Ascomata. (**I**) Section of ascomata. (**J**) Asci. (**K**) Ascospores. Scale bar: (**G**) = 100 μm; (**H**,**I**) = 50 μm; (**J**,**K**) = 10 μm.

**Figure 5 jof-09-00761-f005:**
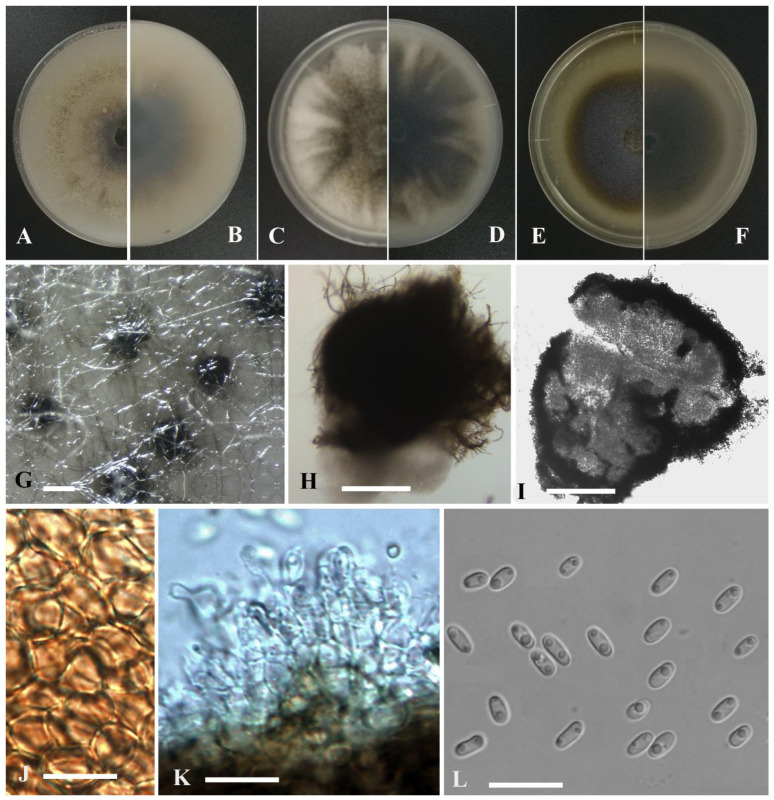
*Stagonosporopsis malaiana* (YMF1.05087). (**A**,**B**) Colony on OA medium (front and reverse) after 7 days. (**C**,**D**) Colony on PDA medium (front and reverse) after 7 days. (**E**,**F**) Colony on MEA medium (front and reverse) after 7 days. (**G**) Pycnidia on OA medium. (**H**) Pycnidia. (**I**) Section of pycnidia. (**J**) Pycnidial wall. (**K**) Conidiogenous cells. (**L**) Conidia. Scale bar: (**G**) = 100 μm; (**H**,**I**) = 50 μm; (**J**,**L**) = 10 μm.

**Figure 6 jof-09-00761-f006:**
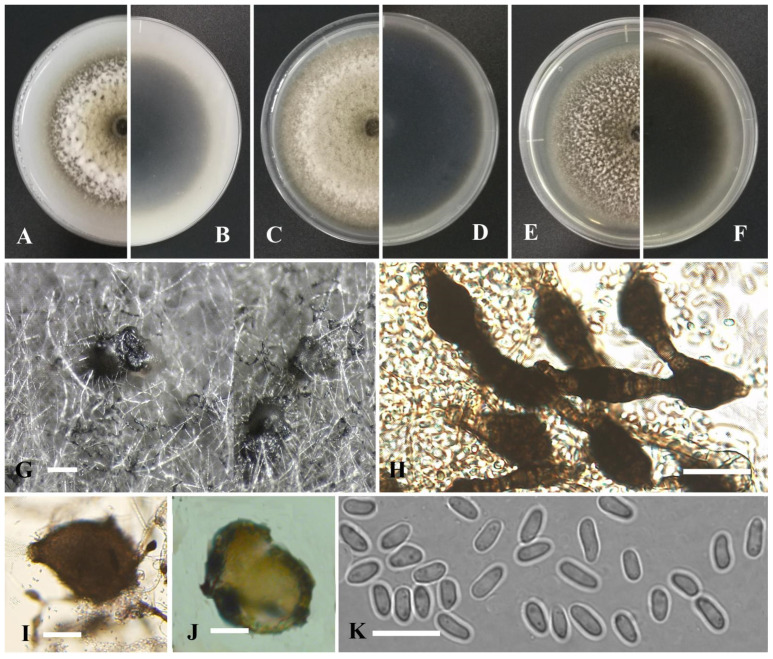
*Didymella hippuris* (YMF1.05089). (**A**,**B**) Colony on OA medium (front and reverse) after 7 days. (**C**,**D**) Colony on PDA medium (front and reverse) after 7 days. (**E**,**F**) Colony on MEA medium (front and reverse) after 7 days. (**G**) Pycnidia on OA medium. (**H**) Chlamydospores. (**I**) Pycnidia. (**J**) Section of pycnidia. (**K**) Conidia. Scale bar: (**G**) = 100 μm; (**I**,**J**) = 50 μm; (**H**,**K**) = 10 μm.

**Figure 7 jof-09-00761-f007:**
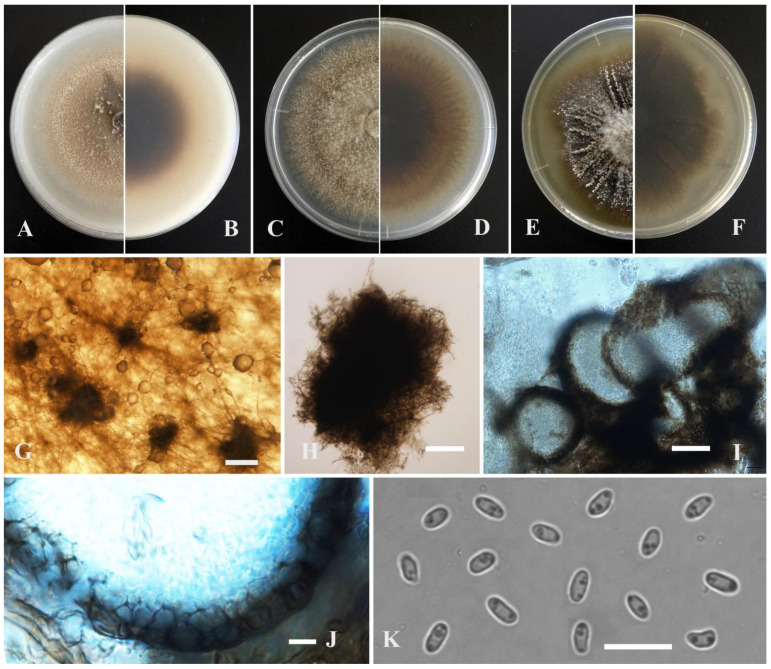
*Didymella erhaiensis* (YMF1.05021). (**A**,**B**) Colony on OA medium (front and reverse) after 7 days. (**C**,**D**) Colony on PDA medium (front and reverse) after 7 days. (**E**,**F**) Colony on MEA medium (front and reverse) after 7 days. (**G**) Pycnidia on OA medium. (**H**) Section of pycnidia. (**I**) Pycnidia. (**J**) Pycnidial wall. (**K**) Conidia. Scale bar: (**G**) = 100 μm; (**H**,**I**) = 50 μm; (**J**,**K**) = 10 μm.

**Figure 8 jof-09-00761-f008:**
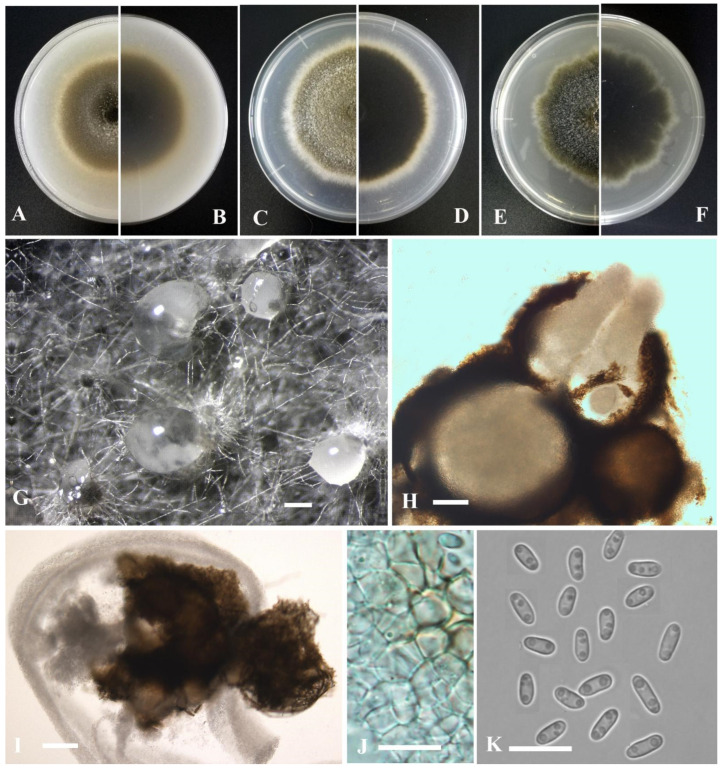
*Didymella myriophyllana* (YMF1.05100). (**A**,**B**) Colony on OA medium (front and reverse) after 7 days. (**C**,**D**) Colony on PDA medium (front and reverse) after 7 days. (**E**,**F**) Colony on MEA medium (front and reverse) after 7 days. (**G**) Pycnidia on OA medium. (**H**) Section of pycnidia. (**I**) Pycnidia. (**J**) Pycnidial wall. (**K**) Conidia. Scale bar: (**G**) = 100 μm; (**H**,**I**) = 50 μm; (**J**,**K**) = 10 μm.

**Figure 9 jof-09-00761-f009:**
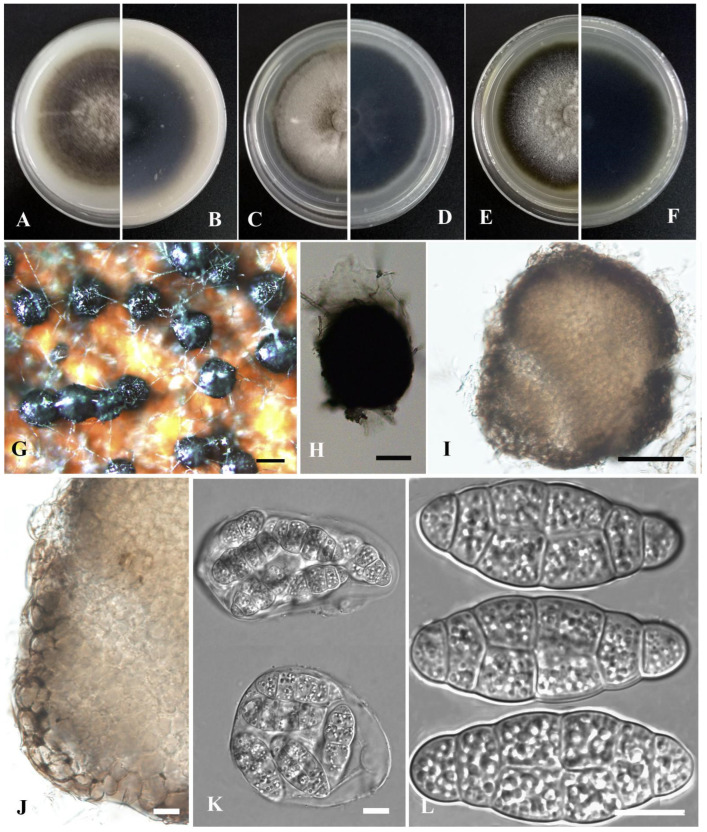
*Leptosphaerulina shangrilensis* (YMF1.05053). (**A**,**B**) Colony on OA medium (front and reverse) after 7 days. (**C**,**D**) Colony on PDA medium (front and reverse) after 7 days. (**E**,**F**) Colony on MEA medium (front and reverse) after 7 days. (**G**) Ascomata on OA medium. (**H**) Ascomata. (**I**) Section of ascomata. (**J**) The wall of ascomata. (**K**) Asci. (**L**) Ascospores. Scale bar: (**G**) = 100 μm; (**H**,**I**) = 50 μm; (**J**,**L**) = 10 μm.

**Figure 10 jof-09-00761-f010:**
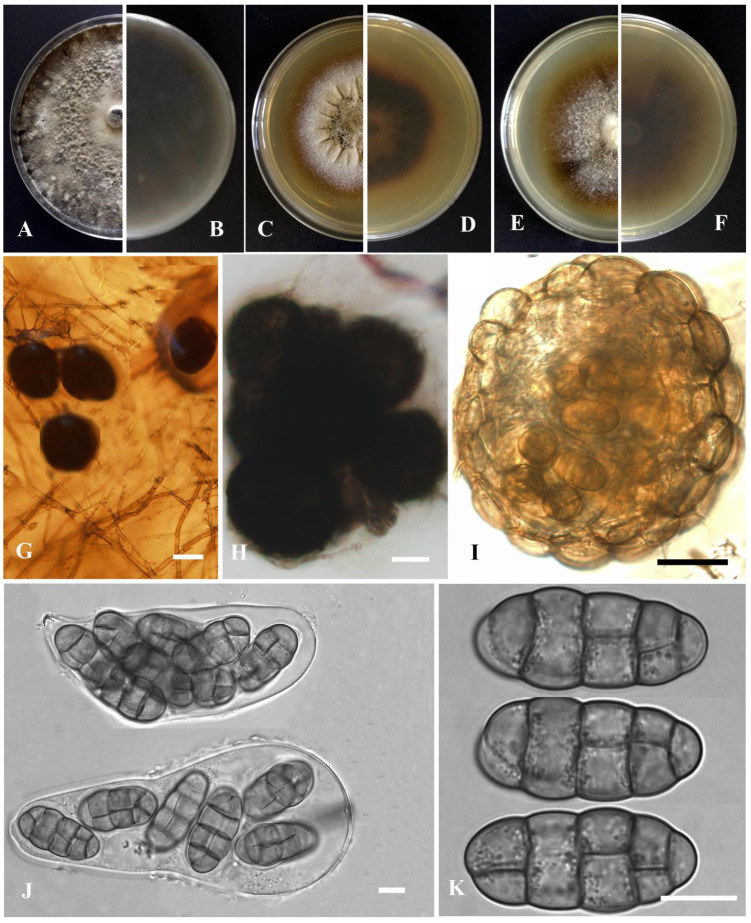
*Leptosphaerulina kasensis* (YMF1.05041). (**A**,**B**) Colony on OA medium (front and reverse) after 7 days. (**C**,**D**) Colony on PDA medium (front and reverse) after 7 days. (**E**,**F**) Colony on MEA medium (front and reverse) after 7 days. (**G**) Ascomata on OA medium. (**H**) Ascomata. (**I**) Section of ascomata. (**J**) Asci. (**K**) Ascospores. Scale bar: (**G**) = 100 μm; (**H**,**I**) = 50 μm; (**J**,**K**) = 10 μm.

**Figure 11 jof-09-00761-f011:**
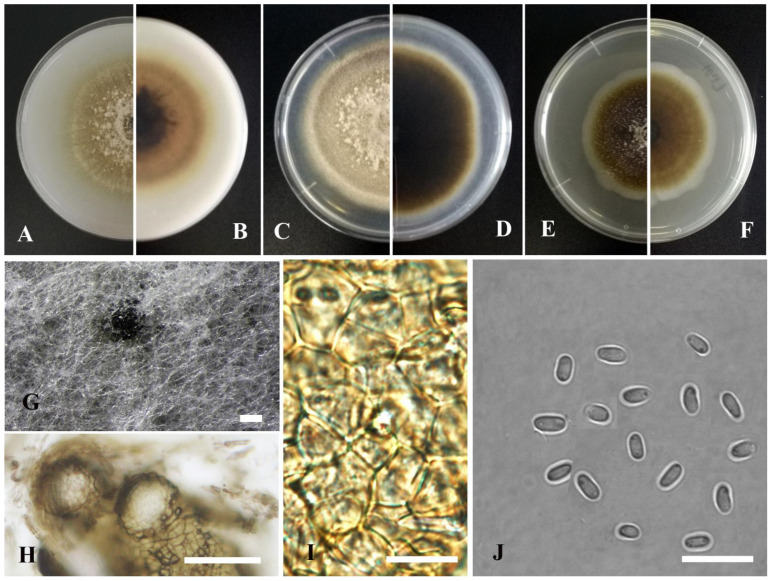
*Cumuliphoma lijiangensis* (YMF1.05096). (**A**,**B**) Colony on OA medium (front and reverse) after 7 days. (**C**,**D**) Colony on PDA medium (front and reverse) after 7 days. (**E**,**F**) Colony on MEA medium (front and reverse) after 7 days. (**G**) Pycnidia on OA medium. (**H**) Section of pycnidia. (**I**) Pycnidial wall. (**J**) Conidia. Scale bar: (**G**) = 100 μm; (**H**) = 50 μm; (**I**,**J**) = 10 μm.

**Figure 12 jof-09-00761-f012:**
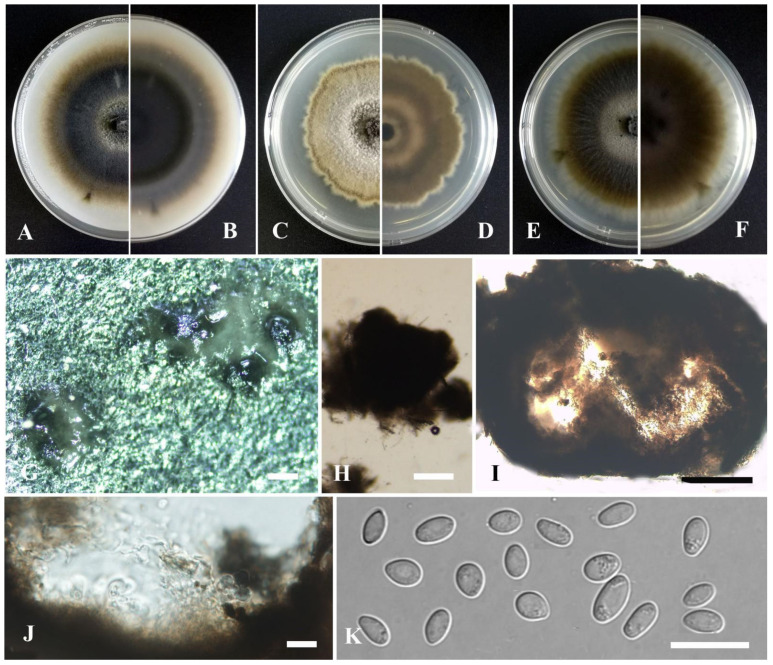
*Dimorphoma isotiana* (YMF1.05048). (**A**,**B**) Colony on OA medium (front and reverse) after 7 days. (**C**,**D**) Colony on PDA medium (front and reverse) after 7 days. (**E**,**F**) Colony on MEA medium (front and reverse) after 7 days. (**G**) Pycnidia on OA medium. (**H**) Pycnidia. (**I**) Section of pycnidia. (**J**) Pycnidial wall. (**K**) Conidia. Scale bar: (**G**) = 100 μm; (**H**,**I**) = 50 μm; (**J**,**K**) = 10 μm.

**Figure 13 jof-09-00761-f013:**
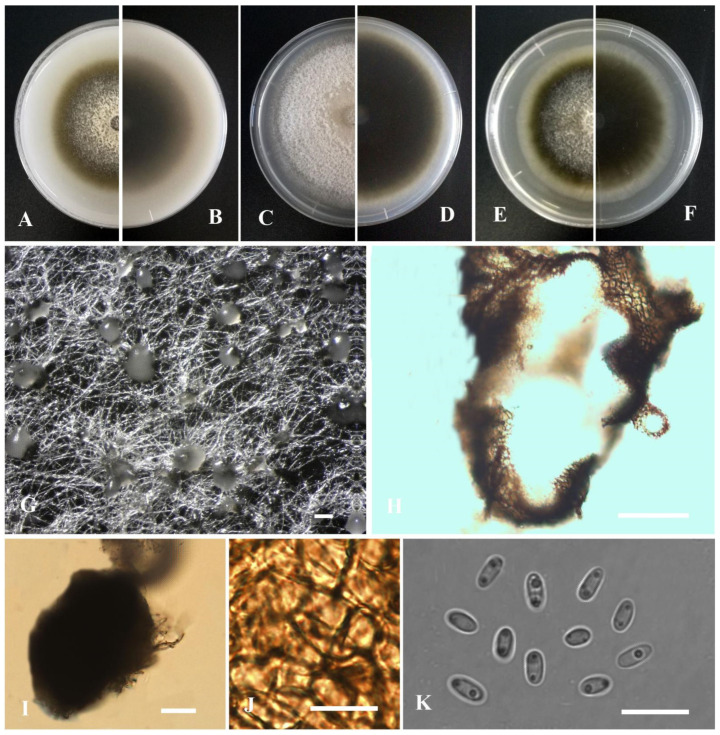
*Didymella gongkaensis* (YMF1.05029). (**A**,**B**) Colony on OA medium (front and reverse) after 7 days. (**C**,**D**) Colony on PDA medium (front and reverse) after 7 days. (**E**,**F**) Colony on MEA medium (front and reverse) after 7 days. (**G**) Pycnidia on OA medium. (**H**) Section of pycnidia. (**I**) Pycnidia. (**J**) Pycnidial wall. (**K**) Conidia. Scale bar: (**G**) = 100 μm; (**H**,**I**) = 50 μm; (**J**,**K**) = 10 μm.

**Figure 14 jof-09-00761-f014:**
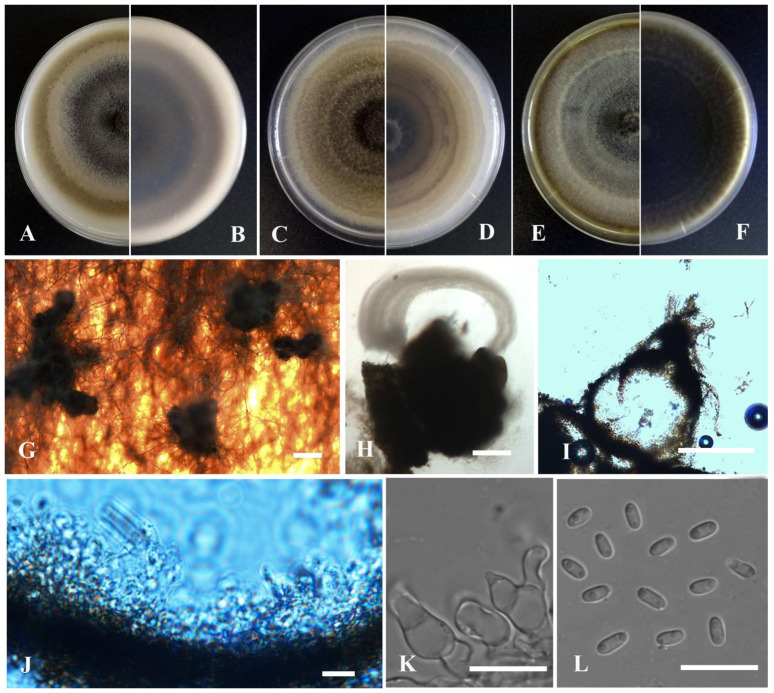
*Ectophoma myriophyllana* (YMF1.05050). (**A**,**B**) Colony on OA medium (front and reverse) after 7 days. (**C**,**D**) Colony on PDA medium (front and reverse) after 7 days. (**E**,**F**) Colony on MEA medium (front and reverse) after 7 days. (**G**) Pycnidia on OA medium. (**H**) Pycnidia. (**I**) Section of pycnidia. (**J**) Pycnidial wall. (**K**) Conidiogenous cells. (**L**) Conidia. Scale bar: (**G**) = 100 μm; (**H**,**I**) = 50 μm; (**J**,**L**) = 10 μm.

**Figure 15 jof-09-00761-f015:**
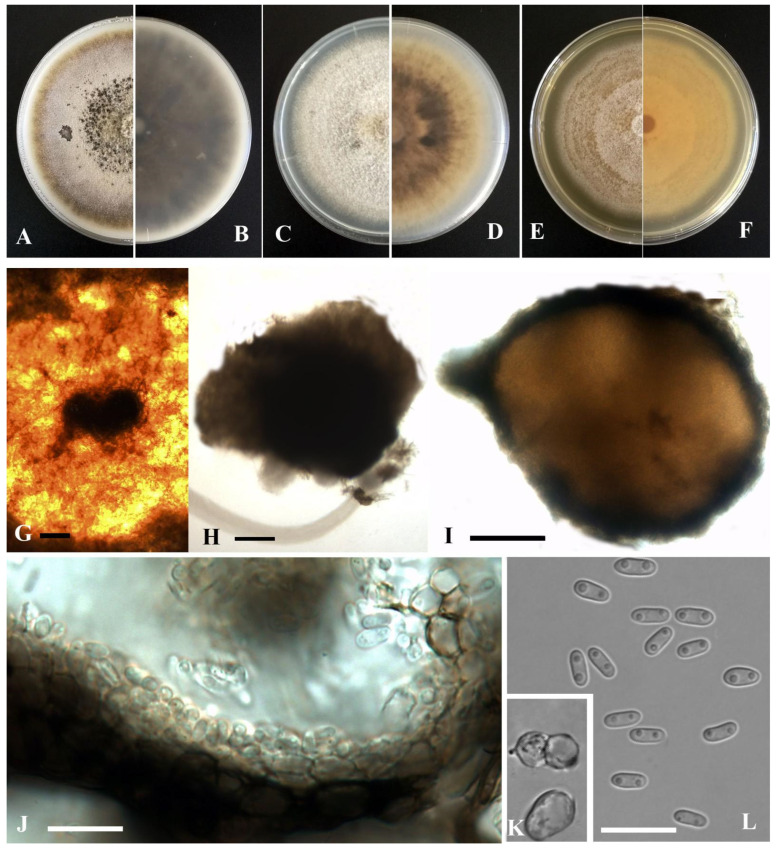
*Remotididymella hydrillana* (Holotype YMF1.05022). (**A**,**B**) Colony on OA medium (front and reverse) after 7 days. (**C**,**D**) Colony on PDA medium (front and reverse) after 7 days. (**E**,**F**) Colony on MEA medium (front and reverse) after 7 days. (**G**) Pycnidia on OA medium. (**H**) Pycnidia. (**I**) Section of pycnidia. (**J**) Pycnidial wall. (**K**) Conidiogenous cells. (**L**) Conidia. Scale bar: (**G**) = 100 μm; (**H**,**I**) = 50 μm; (**J**,**L**) = 10 μm.

## Data Availability

All sequence data are available in NCBI GenBank following the accession numbers listed in the manuscript.

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
