# Peer review of "New Species of Didymellaceae within Aquatic Plants from Southwestern China"

_jof, 2023, doi:10.3390/jof9070761_

Round 1

Reviewer 1 Report

Please find the attachment with some corrections to the manuscript.
The manuscript "New species of Didymellaceae within aquatic plants" describes new fungal species isolated from aquatic plants collected in China.
However, based on the description at the end of the Introduction and Material and Methods, it is challenging to realize whether the presented results are based on previously published or new ones.
The abstract should describe present results not previously published. The introduction requires at least a short paragraph about the aim of the study. Moreover, it would be an excellent addition to include information about the significance of the interactions between aquatic plants and fungi.
The manuscript requires extensive editing, e.g., a space between a word and a parenthesis for all citations.

Minor editing of English language is required.

Author Response

We feel great thanks for your professional review work on our article. As you are concerned, there are several problems that need to be addressed. According to your nice suggestions, we have made extensive corrections to our previous manuscript, the detailed corrections are listed below.

Point 1:

The manuscript "New species of Didymellaceae within aquatic plants" describes new fungal species isolated from aquatic plants collected in China.

However, based on the description at the end of the Introduction and Material and Methods, it is challenging to realize whether the presented results are based on previously published or new ones.

Response 1:

The species of Didymellaceae were studied systematically, and all species have molecular data (Table S1).

After we acquired ITS sequences of our strains, we compared them with known species using BLAST, and got which species were the most phylogenetically close to our strains. Then, we downloaded sequences of these species, and carried out phylogenetic analysis of our strains. Based on the results of analysis, we determined which strains may be new species, and compared them with phylogenetically close species morphologically. Combining phylogenetic analyses and morphological characteristics, we determined new species.

Point 2: The abstract should describe present results not previously published.

Response 2: The abstract was rephrased according to your suggestion.

Point 3: The introduction requires at least a short paragraph about the aim of the study.

Response 3: Thank you for the suggestion. We have added this content in the last paragraph in the introduction.

Point 4: it would be an excellent addition to include information about the significance of the interactions between aquatic plants and fungi.

Response 4: We have added the information in the introduction (Page 2, Line 49-52).

Point 5: The manuscript requires extensive editing, e.g., a space between a word and a parenthesis for all citations.

Response 5: Thank you for your careful examination. We are sorry for our carelessness. According to your comments, we have revised the manuscript.

Point 6: In Abstract, Line 19-20, what about conclusion? what the discovery of new species will mean for modern fungal biology?

Response 6: We have added content which shows the meaning of the discovery of new species for modern fungal biology (Page 1, Line 25-29).

“The high frequency of new species indicates that aquatic plants may be a special ecological niche which highly promote species differentiation. At the same time, the frequent occurrence of new species may indicate the need for extensive investigation of fungal resources in those aquatic environments where fungal diversity may be underestimated.” was added.

Point 7: Page 1, Line 31, these sentences need to be re-written because they sound strange.

Response 7: We have re-written this part according to the suggestion. Page 1, Line 38-40, “Aquatic plants play an important role in maintaining water quality including removing excess nutrient loads, absorbing nutrient mineral ions and reducing sediment resuspension” was added.

Point 8: Page 2, Line 75-81, And what about the present study? What is new addition to your previous discoveries?

Response 8: Present study is exploring new species of Didymellaceae from aquatic plant from southwest of China. Combined with multi-site phylogenetic analysis and morphological characteristics, fungi of Didymellaceae was identified at the species level. In our previous discoveries (Zheng et al. Front. Fungal Biol. 2021, 2: 27, doi: 10.3389/ffunb.2021.692549), fungal strains were identified at the generic level based on ITS.

Point 9: Comments on the Quality of English Language: Minor editing of English language is required.

Response 9: Thanks for your suggestion. We have tried our best to polish language in the revised manuscript.

Reviewer 2 Report

I present my congratulations for the relevant paper  in well-written English and sound presentation. This manuscript presents new and elegant scientific information on fungal taxonomy and diversity that enriches our knowledge on  Didymellaceae in Nature.

I would like to  call your attention for some minor revisions required since I found myself somewhat confused by the sentences below:

In Material and Methods section, 2.1. Collection (lines 82-83 on):

It describes the collection effort, foccusing in the number of areas (24 areas) and plant species (13), but no information is described on the number of samples of each plant species in each area. Also, the habits are pooorly cited. I would suggest an improved description of the real number of samples in this section or when presenting the results of the collections. How many samples in each area, Yunnan, Ghizou and Sichuan? Which area the plant species collected? No list is provided but in the supplementary material. How many samples in freshwater lakes, and in rivers and wetlands? 

Another doubt I was left to deal with is the collection of plants: were the plants collected ramdomly? Or were the individuals chosen by some criteria? Were the plants taken to lab as whole indiviuals or plant parts such as leaves roots flowers, etc? 

In section 2.2 Isolation of Endophytic fungi, line 93: It is described that "each plant sample was cut into 20-20 mm and subsequently (...) cut into small pieces of 0.5cm". I believe there is a mistake in measurements here: Would it be "each plant sample was cut into 20-30 cm"? Since it would be impossible to cut 0.5cm of a 20-30 mm (0.2-0.3cm) piece of plant tissue.

In Results, line 185, it is stated "fifty-one isolates": were these all isolates obtained? Or these isolates are the Didymellaceae only? And other groups were not considered?

In Discussion, line 770: "In our study, 51 isolates from 1,689 taxa" is not clear to me the meaning of taxa here, since 51 isolates do not fit a high figure of 1,689 taxa.

Line 774 "these Didymellaceae (...) rather broad range of hosts generas", please correct to GENERA since this is the plural form of GENUS. Also, the fact that "25 (isolates) were from Hipuris vulgaris and 8 from M. spicatum, with half of the isolates from these two aquatic (host) species" does not prove or point for a rather broad range of host genera. On the contrary, it points for a narrow host range if those numbers represent the majority of isolates. Please explain your conclusion.

Line 776: "Stems and leaves (...) so the environment for fungal colonization (..) is more complex (...). Please, explain what is meant by "rate of colonization" and why this feature points for complexity of the environment.

I belive that the minor revisions suggested besides the description of ecological methods used, would clarify some doubts on the reading.

Best regards

Author Response

Point 1: In Material and Methods section, 2.1. Collection (lines 82-83 on):

It describes the collection effort, foccusing in the number of areas (24 areas) and plant species (13), but no information is described on the number of samples of each plant species in each area. Also, the habits are pooorly cited. I would suggest an improved description of the real number of samples in this section or when presenting the results of the collections. How many samples in each area, Yunnan, Ghizou and Sichuan? Which area the plant species collected? No list is provided but in the supplementary material. How many samples in freshwater lakes, and in rivers and wetlands?

Response: In our previous publication(Zheng et al. Front. Fungal Biol. 2021, 2: 27, doi: 10.3389/ffunb.2021.692549), habits, the results of the collections et al. have been listed in detail. In present study, we mainly focus on new taxon of Didymellaceae, we provided detailed information of type strains. Ecological analyses of aquatic fungi have been published, and this will be repeated if we present them here.

Question 1: No information is described on the number of samples of each plant species in each area. Also, the habits are pooorly cited.

Response 1: We think this is a good suggestion. We added the number of samples of each plant species in each area(at least 15 individuals), and the details see Table S2. The information of habits have been described in previous publication.

Question 2: How many samples in each area, Yunnan, Ghizou and Sichuan?

Response 2: We have added this information. Please see Table S2 for details.

Question 3: Which area the plant species collected?

Response 3: In previous publication, we have listed this information.

Question 4: How many samples in freshwater lakes, and in rivers and wetlands

Response 4: We have added this information. Please see Table S2 for details.

Point 2: Another doubt I was left to deal with is the collection of plants: were the plants collected ramdomly? Or were the individuals chosen by some criteria? Were the plants taken to lab as whole indiviuals or plant parts such as leaves roots flowers, etc?

Response 2: The collected aquatic plants are mainly dominant plants in each sampling site. We collected plants growing near shores, at the same time, because each aquatic plant grow randomly, we collected 15 individuals of the same species at each lake or pond as possible, and there are at least 5-10 meters between two individuls of the same species, not based on quadrat method. Individuals were chosen based on mature and healthy criteria.

The plants taken to lab were as whole individual. After the aquatic plant samples were thoroughly washed through tap water, segments of lamina were taken from the middle portion of fresh healthy leaves, segments from the basal part of the petiole and segments from the fresh root were taken with the help of sterile scissor. For details, see the sections 2.1 and 2.2.

Point 3: In section 2.2 Isolation of Endophytic fungi, line 93: It is described that "each plant sample was cut into 20-30 mm and subsequently (...) cut into small pieces of 0.5cm". I believe there is a mistake in measurements here: Would it be "each plant sample was cut into 20-30 cm"? Since it would be impossible to cut 0.5cm of a 20-30 mm (0.2-0.3cm) piece of plant tissue.

Response 3: We think there is no mistake, we collected whole plant, and washed it using tap water. In order to sterilize surface,whole plant was cut into 20-30 mm, i.e. 2-3cm. After surface sterilization, in order to isolate fungi as possible, samples of 2-3 cm were further cut into small pieces of approximately 0.5 cm length. We have revised this part to make the meaning clearer (Line 114).

Point 4: In Results, line 185, it is stated "fifty-one isolates": were these all isolates obtained? Or these isolates are the Didymellaceae only? And other groups were not considered?

Response 4:

Fifty-one isolates are the Didymellaceae only. Because this study only focuses on new species of the Didymellaceae, other groups were not considered. Present work was based on previous work which identified all strains at genus level based on ITS (published). In this study, we chosen strains of Didymellaceae and identified them at species level.

Point 5: In Discussion, line 770: "In our study, 51 isolates from 1,689 taxa" is not clear to me the meaning of taxa here, since 51 isolates do not fit a high figure of 1,689 taxa.

Response 5:

Based on the results of our previous studies (Zheng et al. Front. Fungal Biol. 2021, 2: 27), the diversity of endophytic fungi of aquatic plants in southwest China was examined, and a total of 1,689 fungal isolates belonging to three phyla and 154 genera were obtained. In the present study, 51 strains of Didymellaceae were selected for further study. Thirty-three of the 51 isolates were identified as 14 new taxa in the family. We have removed “51 isolates from 1,689 taxa” in discussion to make the presented results clearer.

Point 6: Line 774 "these Didymellaceae (...) rather broad range of hosts generas", please correct to GENERA since this is the plural form of GENUS. Also, the fact that "25 (isolates) were from Hipuris vulgaris and 8 from M. spicatum, with half of the isolates from these two aquatic (host) species" does not prove or point for a rather broad range of host genera. On the contrary, it points for a narrow host range if those numbers represent the majority of isolates. Please explain your conclusion.

Response 6:

Thank you for pointing out the inappropriate term usage. We have replaced “generas” with “genera” in the text.

In the present version, we revised point about host specificity (Line 790).

Point 7: Line 776: "Stems and leaves (...) so the environment for fungal colonization (..) is more complex (...). Please, explain what is meant by "rate of colonization" and why this feature points for complexity of the environment.

Response 7: In present version, we move this speculation, and give another one. We speculated that differences of isolation frequence of fungi may be anaerobic environment which resulted isolation frequence of fungi is lower, and see Line 793-795 for details.

Round 2

Reviewer 1 Report

The manuscript "New species of Didymellaceae within aquatic plants" has been improved substantially according to reviewers' comments and suggestions, and it is ready to be published in JoF.